# Limited column formation in the embryonic growth plate implies divergent growth mechanisms during pre- and postnatal bone development

Sarah Rubin[1†], Ankit Agrawal[1,2]*[†], Anne Seewald[1], Meng-Jia Lian[3], Olivia Gottdenker[3], Paul Villoutreix[4], Adrian Baule[5], Tomer Stern[3], Elazar Zelzer[1]*

[1]Department of Molecular Genetics, Weizmann Institute of Science, Rehovot, Israel; [2]Würzburg Institute of Systems Immunology, Julius-Maximilians-Universität Würzburg, Würzburg, Germany; [3]Department of Biologic and Materials & Prosthodontics, University of Michigan School of Dentistry, Ann Arbor, United States; [4]Aix Marseille Univ, INSERM, MMG, UMR1251, Turing Center for Living Systems, Marseille, France; [5]School of Mathematical Sciences, Queen Mary University of London, London, United Kingdom

**\*For correspondence:**
ankitbioinfo@gmail.com (AA);
eli.zelzer@weizmann.ac.il (EZ)

[†]These authors contributed equally to this work

**Competing interest:** The authors declare that no competing interests exist.

**Abstract** Chondrocyte columns, which are a hallmark of growth plate architecture, play a central role in bone elongation. Columns are formed by clonal expansion following rotation of the division plane, resulting in a stack of cells oriented parallel to the growth direction. In this work, we analyzed hundreds of Confetti multicolor clones in growth plates of mouse embryos using a pipeline comprising 3D imaging and algorithms for morphometric analysis. Surprisingly, analysis of the elevation angles between neighboring pairs of cells revealed that most cells did not display the typical stacking pattern associated with column formation, implying incomplete rotation of the division plane. Morphological analysis revealed that although embryonic clones were elongated, they formed clusters oriented perpendicular to the growth direction. Analysis of growth plates of postnatal mice revealed both complex columns, composed of ordered and disordered cell stacks, and small, disorganized clusters located in the outer edges. Finally, correlation between the temporal dynamics of the ratios between clusters and columns and between bone elongation and expansion suggests that clusters may promote expansion, whereas columns support elongation. Overall, our findings support the idea that modulations of division plane rotation of proliferating chondrocytes determines the formation of either clusters or columns, a multifunctional design that regulates morphogenesis throughout pre- and postnatal bone growth. Broadly, this work provides a new understanding of the cellular mechanisms underlying growth plate activity and bone elongation during development.

## Editor's evaluation

The study presents a landmark finding on quantifying the orientation and organization of chondrocyte columns in the prenatal and postnatal growth plate cartilage using advanced 3D imaging and a sophisticated image analysis pipeline. The evidence supporting the authors' conclusions regarding clusters of chondrocytes (instead of columns) in the fetal growth plate is considered compelling, with rigorous imaging analyses. The work will be of broad interest to developmental and cell biologists.

**eLife digest** As we develop, the long bones in our arms and legs must grow bigger and stronger to support our weight and movements. The width and length of these bones increase rapidly while in the womb, but after birth, they lengthen more quickly than they widen.

Both expansion and extension occur at the growth plates, two narrow zones located at each bone's ends and which host cells that can divide and increase in size. Traditionally, bone lengthening has been understood resulting from these 'chondrocytes' expanding in size after having organized themselves into columns that run parallel to the long axis of the bone. This is possible due to newly born cells performing a complex 90-degree rotation that results in this characteristic organization in column stacks. How bones widen, however, is less well-understood.

To shed light on these mechanisms, Rubin, Agrawal et al. took advantage of recent technologies that allowed them to track the spatial organization of cells in 3D during development. Their experiments showed that, in mice, chondrocytes in the growth plate were rarely organized in columns before birth, with most cells not performing a 90-degree rotation of their division plane. This led to most clusters growing perpendicularly to the long axis of the bone, resulting in bone widening.

After birth, however, most chondrocytes successfully completed the rotation, establishing columns running parallel to the long axis; fewer clusters contributing to the widening of the bone were present.

Taken together, these results suggest that controlling the rotation of the division plane in chondrocytes helps create different growth strategies before and after birth. They also indicate that elongation in the womb may not require chondrocytes to be systematically organized in columns. Overall, the findings by Rubin, Agrawal et al. point to new mechanisms underpinning bone growth, which could be important to investigate further in both health and disease.

## Introduction

Cellular organization plays a major role in tissue and organ morphogenesis (*Lecuit and Le Goff, 2007*; *Lecuit and Lenne, 2007*; *Irvine and Wieschaus, 1994*; *Bailles et al., 2022*; *Collinet and Lecuit, 2021*; *Sutherland et al., 2020*). The mammalian growth plate is an excellent example for this concept as its complex architecture is the engine driving longitudinal bone growth (*Rubin et al., 2021*; *Breur et al., 1991*; *Wilsman et al., 2008*; *Wilsman et al., 1996*; *Cooper et al., 2013*). The growth plate, which is located at both ends of developing long bones, drives bone elongation by a tightly regulated process of cell proliferation and differentiation, which involves increase in cell size and their organization along the proximal-distal (P-D) axis (*Kronenberg, 2003*; *Mackie et al., 2008*; *Cancedda and Cancedda, 1995*; *Noonan et al., 1998*). The growth plate comprises four zones. At the most distal epiphyseal end is the resting zone (RZ), where chondrocytes are small and disorganized. Underneath lies the proliferative zone (PZ), where chondrocytes increase in volume, adopt a flat and elongated morphology, and organize into columns (*Abad et al., 2002*; *Dodds, 1930*; *Li and Dudley, 2009*; *Li et al., 2017*; *Romereim et al., 2014*). In the subsequent prehypertrophic (PHZ) and hypertrophic zones (HZ), cells reach their maximum size (*Rubin et al., 2021*; *Cooper et al., 2013*; *Breur et al., 1997*). These changes in cell size and spatial organization determine the rate of bone elongation (*Breur et al., 1991*; *Wilsman et al., 2008*; *Wilsman et al., 1996*; *Kember and Walker, 1971*; *Lui et al., 2018*; *Li et al., 2015*).

Columnar arrangement of chondrocytes has been a subject of study for nearly a century (*Dodds, 1930*), gaining attention due to the remarkable emergence of cellular order from the highly disordered RZ. This columnar arrangement facilitates bone elongation by maximizing cell density in the longitudinal axis while limiting it laterally, thereby constraining hypertrophic cell growth to the P-D axis (*Romereim and Dudley, 2011*). In the PZ, the division of column-forming cells is perpendicular to the P-D axis. Considering that these cells ultimately orient themselves with their short axis parallel to the P-D axis, the rearrangement into elongated columns requires a robust morphogenetic mechanism. Originally, analyses of two-dimensional static images suggested that in the embryonic growth plate, columns form through a process akin to convergent extension, an evolutionarily conserved tissue elongation mechanism involving cell intercalation (*Li and Dudley, 2009*; *Ahrens et al., 2009*; *Shwartz et al., 2012*; *Gao et al., 2011*; *Yang et al., 2003*). However, more recent live imaging studies in various model systems showed that cells do not intercalate to form columns (*Li et al., 2017*;

*Romereim and Dudley, 2011*; *Yuan et al., 2023*). Instead, following cell division, sister cells undergo a cell–cell and cell–extracellular matrix (ECM) adhesion-dependent 90° rotation prior to separation. This rotation ensures that cells are neatly stacked with their short axis parallel to the P-D axis.

Recent studies have highlighted three fundamental principles governing column formation. First, columns consist of clonal cells (*Li and Dudley, 2009*; *Li et al., 2017*; *Romereim et al., 2014*; *Ahrens et al., 2009*; *Newton et al., 2019*; *Mizuhashi et al., 2018*; *Hallett et al., 2022*). Whereas embryonic columns are multiclonal, postnatally, following the formation of secondary ossification centers, columns become monoclonal and originate from Pthrp + RZ cells (*Newton et al., 2019*; *Mizuhashi et al., 2018*; *Hallett et al., 2022*). The second principle is that cells within the column orient their short axis parallel to the P-D axis of the bone (*Rubin et al., 2021*; *Li and Dudley, 2009*; *Li et al., 2017*; *Romereim et al., 2014*; *Ahrens et al., 2009*; *Shwartz et al., 2012*; *Aszodi et al., 2003*) within a threshold of 12° (*Li et al., 2017*). The third rule pertains to the alignment of the column itself. The long axis is oriented parallel to the P-D axis (*Dodds, 1930*; *Li and Dudley, 2009*; *Romereim et al., 2014*; *Ahrens et al., 2009*; *Shwartz et al., 2012*; *Aszodi et al., 2003*; *Moss-Salentijn et al., 1987*) within a 12° threshold for single columns and a 20° threshold for complex columns (*Li et al., 2017*).

Over the years, numerous studies have been dedicated to deciphering the molecular and cellular processes underpinning the formation of columns and their involvement in bone elongation. Studies in embryonic and postnatal mouse limbs have shown the importance of interactions between chondrocytes and the surrounding ECM. These studies have identified beta 1 and alpha 10 integrins, along with α-parvin, as physical regulators governing cell polarity and rotation during column formation (*Yuan et al., 2023*; *Aszodi et al., 2003*; *Bengtsson et al., 2005*). Furthermore, studies in embryonic chick and mouse limbs have shown that cell surface signaling through the Fz/Vangl/PCP pathway plays a major role in regulating chondrocyte polarity and rearrangement (*Li and Dudley, 2009*; *Li et al., 2017*; *Ahrens et al., 2009*; *Gao et al., 2011*; *Yang et al., 2003*; *Yang and Mlodzik, 2015*) and that GDF5 is involved in chondrocyte orientation (*Rubin et al., 2021*). Finally, studies in paralyzed mice (*Killion et al., 2017*) and muscle-less mouse embryos *Shwartz et al., 2012*; *Pierantoni et al., 2021* have uncovered the important role of muscle load in regulating cell polarity and column formation.

In this study, we analyze the 3D architecture of confetti-labeled clones in the embryonic and postnatal growth plate of mice. Intriguingly, we found that chondrocytes in the embryonic growth plate are rarely arranged in columns. Instead, successive incomplete rotations during cell division result in non-stereotypic cell stacking that, in turn, give rise to elongated clusters oriented orthogonally to the longitudinal bone axis. However, in the postnatal growth plate clones, the rate of complete cell rotations increases, leading to the formation of complex columns through a combination of stereotypical and non-stereotypic cell stacking, as well as of small, orthogonally oriented clusters. Additionally, we observed that column formation is buffered, permitting deviations of up to 60% incomplete rotations between successive cells within columns. The presence of clusters and columns correlated temporally with the rates of growth plate elongation and expansion, suggesting that these structures support different growth strategies during embryonic and postnatal bone development, while highlighting the imperative role of 3D analysis when studying complex cellular arrangements.

## Results
### 3D imaging of cell clones in the embryonic growth plate reveals stacking patterns that do not support column formation

To date, a comprehensive 3D analysis of column formation in the mouse embryonic growth plate has not been performed. To address this gap, we conducted multicolor clonal lineage tracing on proximal tibia and distal femur growth plates of Col2a1-CreER:R26R-Confetti embryos. Labeled cellular clones were subjected to 3D morphometric analysis using our previously reported 3D MAPs pipeline (*Rubin et al., 2021*; *Figure 1—figure supplement 1*). Recombination was induced at E14.5 and 4 days later, chondrocyte clones were observed in all growth plate zones (*Figure 1A–C*). In 2D optical sections from both femur and tibia, cells within the clones appeared neatly stacked, forming a column-like structure parallel to the P-D bone axis (*Figure 1B and C*). However, 3D examination revealed no columnar organization and cells that were rarely stacked neatly (*Figure 1D and E*, *Figure 1—figure supplement 2*, and *Figure 1—videos 1–11*).

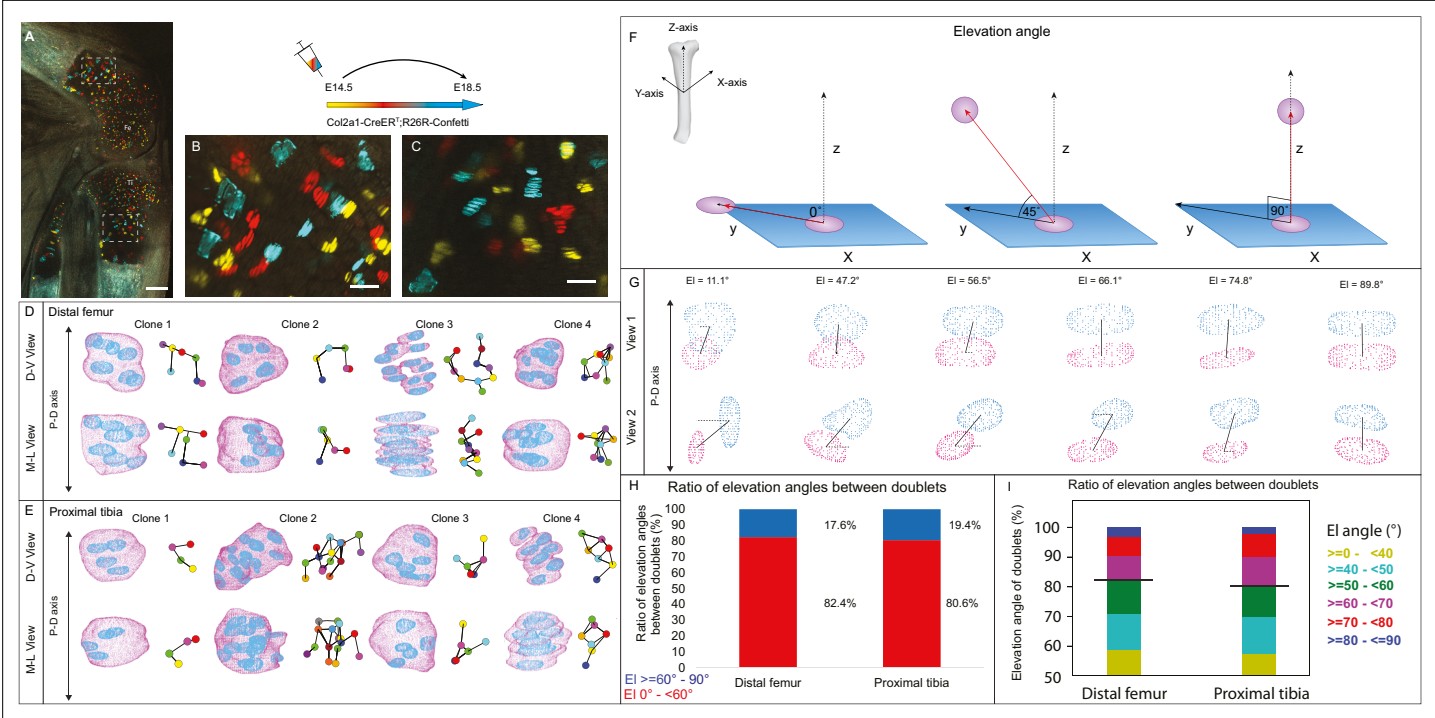

**Figure 1.** 3D imaging of clones in the embryonic growth plate reveals complex morphologies. Chondrocyte clones in the proximal tibia and distal femur growth plates of Col2a1-CreER[T2]:R26R-Confetti heterozygous mice were pulsed by tamoxifen administration at E14.5 and imaged at E18.5. (**A–C**) An image of chondrocyte clones in the knee was captured with a combination of multiphoton and confocal imaging using a Leica TCS SP8 confocal laser-scanning/MP microscope. Sparse labeling is observed throughout the growth plate. Scale bar: 250 μm. Magnified optical section of distal femur (**B**) and proximal tibia (**C**) highlight clones in the proliferative, prehypertrophic and hypertrophic zones, which appear to form columns. Scale bars: 50 μm. (**D, E**) 3D rendering of representative clones along the D-V and M-L axes from the distal femur (**D**) and proximal tibia (**E**) growth plates. Clone surface is in magenta and nuclear surface in blue. Skeletonized illustrations on the right highlight the complexity of clonal morphologies. Nuclear centroids are depicted as randomly colored circles; lines connect between nearest neighbor nuclei. (**F**) Illustration of various elevation angles between the centroids of two cells. An elevation angle of 0° indicates that the two cells are in the same equatorial plane (XY), whereas an elevation angle of 90° indicates that the cells are directly above the equatorial plane in the same XZ plane (perpendicular to the XY plane). The Z axis represents the P-D bone axis. The equatorial plane XY is perpendicular to the Z axis. The red line is the projection of the cell. (**G**) Representative images of nuclei at different elevation angles in two orthogonal viewing angles. Solid black lines represent the shortest distance between nuclear centroids. Elevation angle is the angle between the dashed black line and solid black line. (**H, I**) Stacked histograms show quantification of elevation angles between doublet cells in distal femur (n = 1044) and proximal tibia clones (n = 805). (**H**) Proportion of complete rotations (i.e., elevation angles of 60–90°, in blue) vs incomplete rotations (under 60°, in red). (**I**) Distribution of elevation angles (°) is color-coded as indicated. Black line marks the 60° cutoff. Three biologically independent samples were examined in nine independent experiments.

The online version of this article includes the following video and figure supplement(s) for figure 1:

**Figure supplement 1.** Schematic of experimental workflow and segmentation.

**Figure supplement 2.** Orthogonal viewing angles of raw imaging data.

**Figure 1—video 1.** Volumetric rendering of embryonic clone in the proximal tibia growth plate.
https://elifesciences.org/articles/95289/figures#fig1video1

**Figure 1—video 2.** Volumetric rendering of embryonic clone in the proximal tibia growth plate.
https://elifesciences.org/articles/95289/figures#fig1video2

**Figure 1—video 3.** Volumetric rendering of embryonic clone in the proximal tibia growth plate.
https://elifesciences.org/articles/95289/figures#fig1video3

**Figure 1—video 4.** Volumetric rendering of embryonic clone in the distal femur growth plate.
https://elifesciences.org/articles/95289/figures#fig1video4

**Figure 1—video 5.** Volumetric rendering of embryonic clone in the distal femur growth plate.
https://elifesciences.org/articles/95289/figures#fig1video5

**Figure 1—video 6.** Volumetric rendering of embryonic clone in the distal femur growth plate.
https://elifesciences.org/articles/95289/figures#fig1video6

*Figure 1 continued on next page*

*Figure 1 continued*

**Figure 1—video 7.** Volumetric rendering of embryonic clone in the distal femur growth plate.
https://elifesciences.org/articles/95289/figures#fig1video7

**Figure 1—video 8.** Volumetric rendering of embryonic clone in the distal femur growth plate.
https://elifesciences.org/articles/95289/figures#fig1video8

**Figure 1—video 9.** Volumetric rendering of embryonic clone in the distal femur growth plate.
https://elifesciences.org/articles/95289/figures#fig1video9

**Figure 1—video 10.** Volumetric rendering of embryonic clone in the distal femur growth plate.
https://elifesciences.org/articles/95289/figures#fig1video10

**Figure 1—video 11.** Volumetric rendering of embryonic clone in the distal femur growth plate.
https://elifesciences.org/articles/95289/figures#fig1video11

To characterize the stacking behavior of embryonic growth plate cells, we performed quantitative analysis of local cell stacking in distinct clones. For that, we measured the elevation angle between all neighboring pairs of cells (doublets) within a given clone, similar to what was done previously (*Carolyn, 2023*) (see details in 'Materials and methods' and *Figure 1F*). In a spherical coordinate system, the elevation angle between two perfectly stacked cells at origin 0 would be 90° (*Figure 1F*), indicating a complete rotation of the division plane during cell division. Visualization of cell doublets representative of various elevation angles employing two orthogonal viewing directions revealed that in 3D, typically stacked column cells are expected to exhibit elevation angles in the range of 60–90° (*Figure 1G*). Notably, however, quantification of elevation angles revealed that less than 20% of doublets within a clone were typically stacked (distal femur, 17.6%; proximal tibia, 19.4%; *Figure 1H and I*). These results show that clone cells in the embryonic growth plate do not display the typical stacking behavior associated with a columnar arrangement, as most cells undergo incomplete division plane rotation.

## Columns are rare in the embryonic growth plate

Our finding that embryonic clones did not exhibit typical cell stacking characteristic of columns raised the question of their contribution to bone elongation. Atypically stacked cells, with elevation angles less than 60°, could still support longitudinal growth if the clones have elongated morphologies along the P-D axis of the growth plate. Thus, to characterize clone morphology, we extracted the long, medium, and short axes and measured the ratios between them (*Figure 2A*; see 'Materials and methods'). Ratios close to 1 across all axes would indicate a spherical shape, whereas ratios approaching 0 would reveal a flattened ellipsoidal shape. Results showed that the long axis of the clones was consistently at least twice the length of the short axis. Moreover, in half of the clones the long axes measured at least five times the length of the short axes (distal femur, 58%; proximal tibia, 47%), indicating ellipsoid morphologies (*Figure 2C*). Further examination showed that in roughly half of the clones, the long axis was at least twice the length of the medium axis (distal femur, 58.4%; proximal tibia, 54.3%; *Figure 2B*) and the medium axis was at least twice the length of the short axis (distal femur, 58.5%; proximal tibia, 50.4%; *Figure 2D*). Altogether, these results indicate that embryonic clones are either lentil-shaped oblate ellipsoids or rugby ball-like prolate ellipsoids and, thus, may contribute to bone elongation.

Next, we sought to determine whether the elongated embryonic clones are aligned with the P-D bone axis. Previous studies suggested that in a 2D Cartesian coordinate system, single columns orient their long axis within 12° of the P-D bone axis, whereas multicolumns, that is, those composed of multiple cell stacks, orient within 20° (*Li et al., 2017*). In a 3D spherical coordinate system, these values correspond to elevation angles of 78° and 70°, respectively. We therefore set a more permissive threshold of 60° elevation to determine whether or not a clone qualifies as a column (*Figure 2E*). Measurements of the angle between the long axis of the clone and the P-D bone axis (see 'Materials and methods') revealed that nearly all clones in the proximal tibia (mean, 95.4%) and the distal femur (mean, 97.3%) oriented perpendicular to the P-D axis (*Figure 2F and G*, *Figure 2—figure supplement 1B–D*). On average, only 4.6% of clones in the proximal tibia and 2.7% in the distal femur displayed a column-like orientation. (*Figure 2F and G*, *Figure 2—figure supplement 1C*). Together,

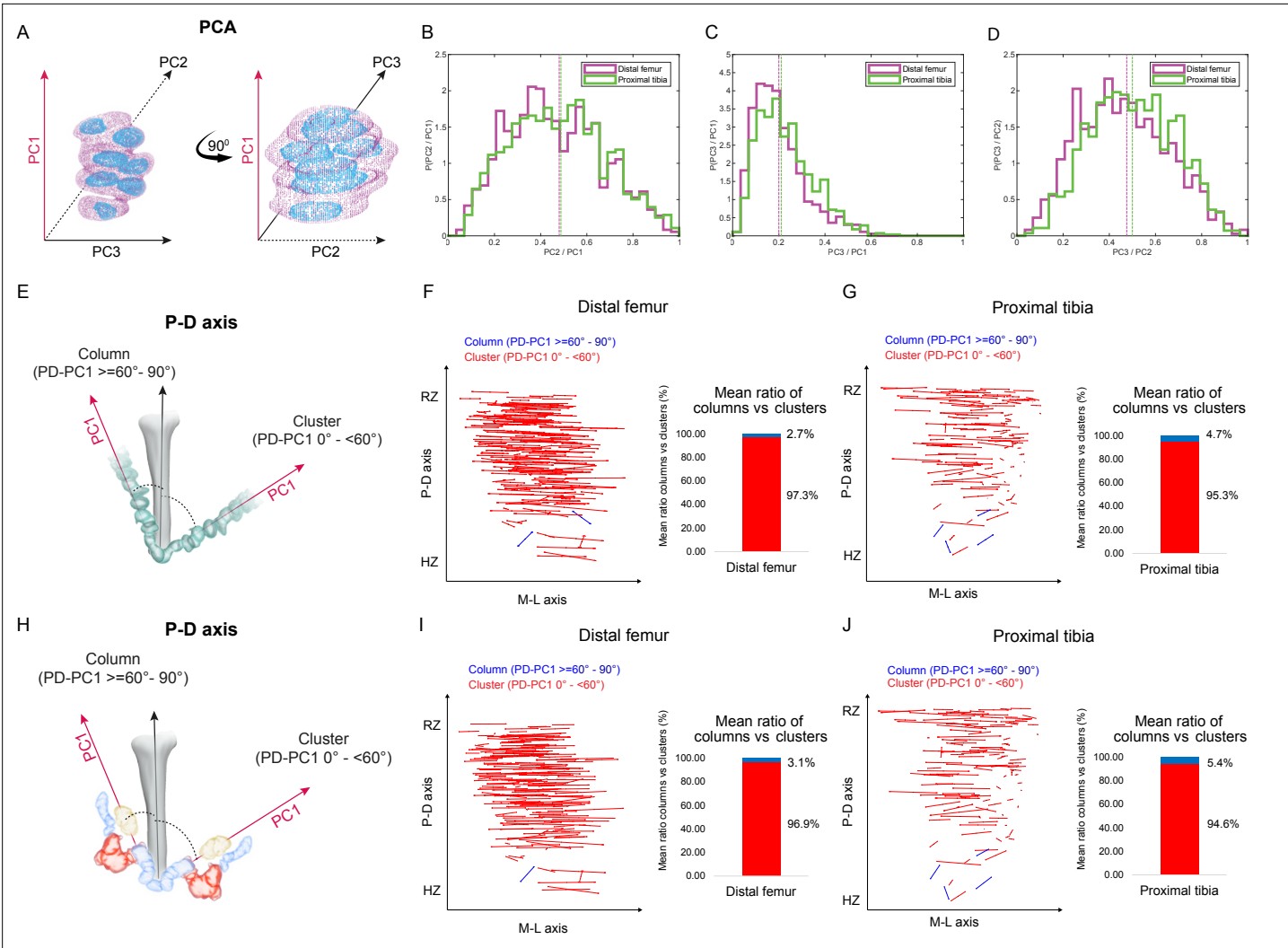

**Figure 2.** Columns are rare in the embryonic growth plate. Clone morphology was extracted by calculating the three orthogonal axes of each clone using principal component analysis (PCA). PC1 (pink arrow). represents the long axis of the clone, PC2 (dashed black arrow) the medium axis, and PC3 (solid black arrow) the short axis. (**A**) A schematic drawing of the same clone from two orthogonal viewing angles with principal components labeled. (**B–D**) Histograms of clone PC ratios in E18.5 distal femur (DF) and proximal tibia (PT) growth plates reveal that clonal morphology is either oblate or prolate ellipsoid. In half of the clones, the long axis was at least twice the size of the medium axis (PC2/PC1: DF mean ± SD, 0.464 ± 0.197; PT, 0.469 ± 0.201; **B**), the long axis was at least five times the size of the short axis (PC3/PC1: DF, 0.201 ± 0.109; PT, 0.226 ± 0.119; **C**), and the medium axis was at least twice the size of the short axis (PC3/PC2: DF, 0.456 ± 0.189; PT, 0.504 ± 0.182; **D**). Dashed lines show the mean between samples. (**E**) Scheme illustrating the threshold between uniclones considered as columns (i.e., angle between long axis of the clone and P-D axis of the bone is 60–90°) or clusters (i.e., angle is below 60°). (**F, G**) Orientation maps along the P-D and M-L axes and quantification of mean ratio of columns (in blue) vs clusters (in red) per sample for clones in the DF (n = 1044; **F**) and PT (n = 805; **G**) growth plates. Each line in the map represents the orientation of the long axis of an individual clone, whereas the length of the line is proportional to that of the clonal long axis (PC1). RZ refers to the middle of the resting zone; HZ refers to the end of the hypertrophic zone. (**H**) Scheme illustrating the threshold between multiclones considered as columns or clusters. (**I, J**) Orientation maps and quantification of mean ratio of multiclonal columns vs clusters per sample for the DF (n = 816; **I**) and PT (n = 619; **J**) growth plates.

The online version of this article includes the following figure supplement(s) for figure 2:

**Figure supplement 1.** Orientation maps and distribution of uniclones in embryonic growth plates.

**Figure supplement 2.** Orientation maps and distribution of multiclones in embryonic growth plates.

these results show that while embryonic clones have elongated morphologies, they do not support longitudinal growth.

## Multiclonal columns are rare in the growth plate

Previous studies have shown that embryonic columns may be formed by merging of multiple clones (*Li et al., 2017*; *Newton et al., 2019*). This opens the possibility that in the embryo columns are multiclonal. To examine this possibility, we allowed neighboring clones to join (see 'Materials and methods') and then performed orientation analysis, measuring the angle between the long axis of the multiclone with the P-D bone axis as before (*Figure 2H*). As depicted in *Figure 2* and *Figure 2—figure supplement 2*, nearly all multiclones in the proximal tibia (mean, 94.6%) and distal femur (mean, 96.8%) were oriented perpendicular to the P-D axis (*Figure 2I and J*, *Figure 2—figure supplement 2A–D*). On average, only 5.37% of multiclones in the proximal tibia and 3.12% in the distal femur were aligned

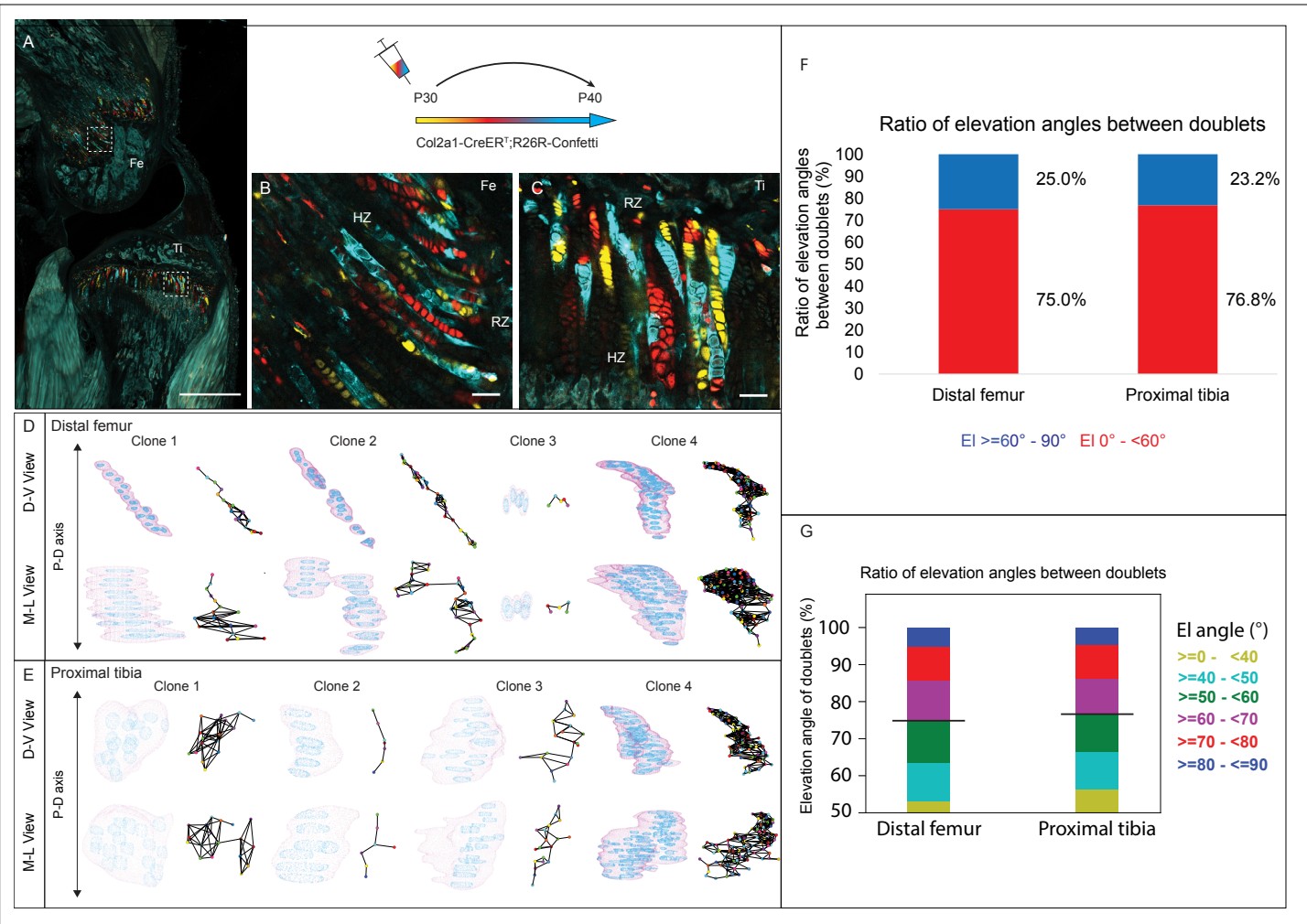

**Figure 3.** 3D imaging of clones in the postnatal growth plate reveals diverse and complex morphologies. 3D morphology of chondrocyte clones was analyzed in the proximal tibia (PT) and distal femur (DF) growth plates of Col2a1-CreER$^{T2}$:R26R-Confetti mice. Cells were pulsed by tamoxifen administration at P30 and traced until P40. (**A**) An image of chondrocyte clones in a P40 mouse knee was captured with a combination of multiphoton and confocal imaging using a Leica TCS SP8 confocal laser-scanning/MP microscope. Sparse labeling is observed throughout the growth plate. Scale bar: 1 mm. (**B, C**) Magnified optical sections of DF and PT clones reveal complex clones that appear to form columns. Scale bars: 50 μm. (**D, E**) 3D rendering of representative clones along the D-V and M-L axes of the DF and PT growth plates. Clone surface is in magenta and nuclear surfaces in blue. Skeletonized illustrations on the right highlight the complexity of each clone. Nuclear centroids are depicted as a randomly colored circle; lines represent connections with nearest neighbor nuclei. (**F, G**) Stacked histograms show quantification of elevation angles between cell doublets in clones. (**F**) Ratio between good rotations (El, 60–90° in blue) and incomplete rotations (El, 0–60°, in red). (**G**) Distribution of elevation angles (°), color-coded as indicated. Black line marks the 60° cutoff. DF, 1866 clones; PT, 1666 clones. RZ, resting zone; HZ, hypertrophic zone; El, elevation angle.

parallel to the P-D bone axis, thereby satisfying the global orientation criterion for a column (*Figure 2I and J*). Together, these results show that embryonically multiclonal columns are rare.

### 3D imaging of postnatal growth plate clones reveals diverse complex morphologies

Having found that embryonic clones do not meet the criteria for columns, we proceeded to analyze the 3D clonal structure in postnatal growth plates. For that, we used the same pipeline (*Rubin et al., 2021*) to analyze proximal tibia and distal femur growth plates from clonally labeled Col2a1-CreER:R26R-Confetti mice (*Figure 1—figure supplement 1A*) at P40, 10 days after Cre induction. 2D optical sections showed small clones in the RZ directly beneath the secondary ossification center, alongside longitudinal clones that spanned most of the growth plate height (*Figure 3A–C*). However, 3D rendering of confetti clones coupled with maps, generated by using multiple viewing angles, highlighted the diversity and complexity of these clones (*Figure 3D and E*). Whereas the small clones expanded along the D-V and M-L axes (*Figure 3D*), the longitudinal axis of most large clones visually aligned with the P-D bone axis. Surprisingly, however, nearly all the large clones, which contained 15–100 cells, had a complex morphology that was apparent from a particular viewing angle. Additionally, each large clone displayed motifs along its length, where cells appeared to stack typically before branching off into a horizontal expansion.

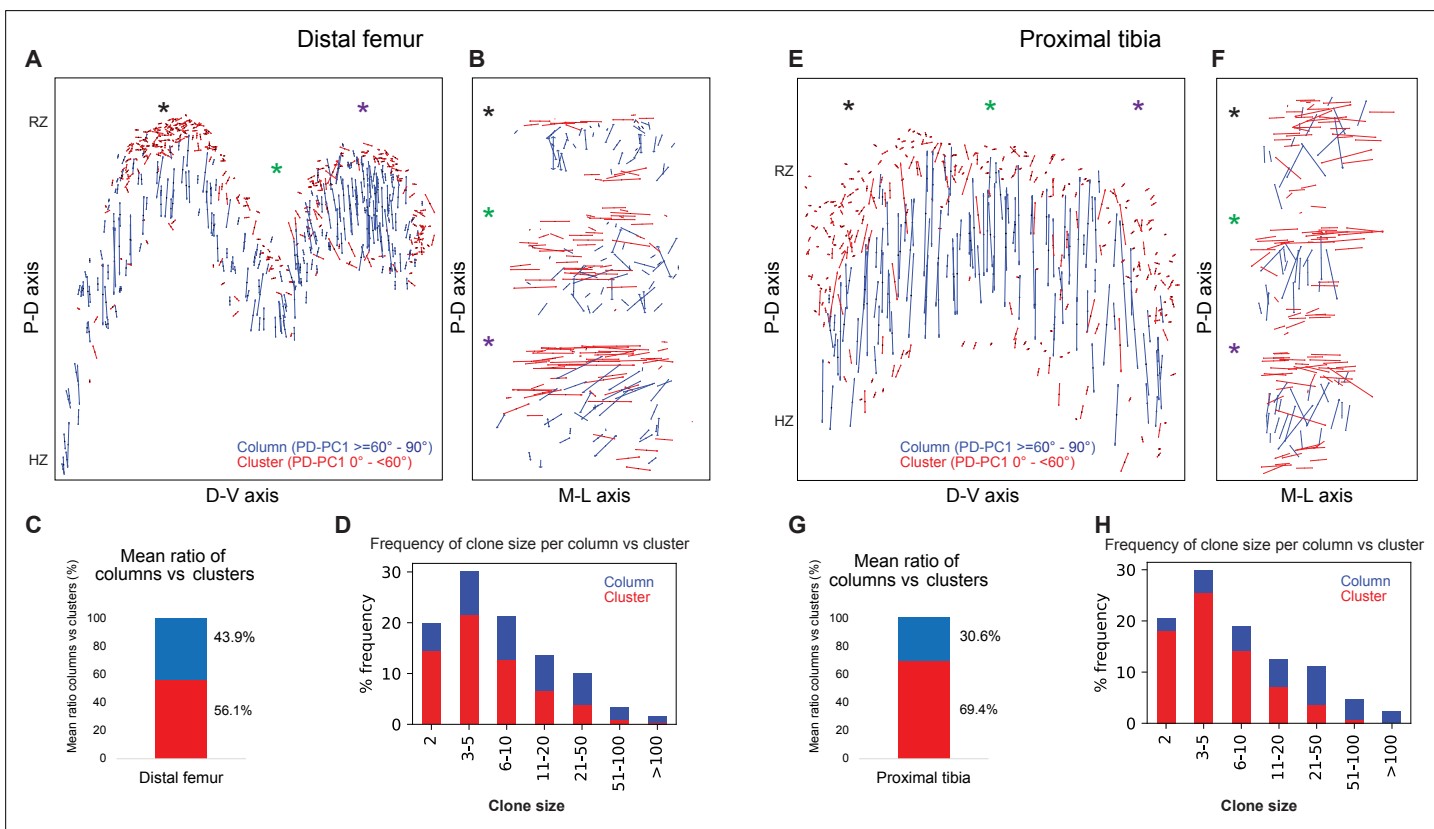

**Figure 4.** Complex longitudinal clones function as columns in the postnatal growth plate. Orientation maps of clones in P40 growth plates. (**A, B**) Clone orientation along the P-D and D-V axes of distal femur (DF) growth plates (n = 737 columns, 1129 clusters). Asterisks indicate the same locations in the growth plate. Each line represents the long axis of an individual clone, with its length proportional to that of the clone long axis. Columns are shown in blue and clusters in red. (**C**) Quantification of mean ratio of columns vs clusters in the DF growth plates. (**D**) Frequency of clone size per column (blue) vs cluster (red). (**E, F**) Clone orientation in the proximal tibia (PT) growth plates (n = 512 columns, 1154 clusters). (**G, H**) Mean ratio of columns vs clusters and frequency of clone size in PT growth plates. RZ, resting zone; HZ, hypertrophic zone. Three biologically independent samples were examined in nine independent experiments.

The online version of this article includes the following figure supplement(s) for figure 4:

**Figure supplement 1.** Clones orientation in the postnatal growth plate.

## Postnatal clones lack stereotypical cell stacking

Next, to determine whether postnatal clones form columns, we examined the two criteria for columns, namely local cell stacking and global orientation parallel to the P-D axis. To determine the degree of local cell stacking in the postnatal clones, we analyzed the elevation angle between all pairs of cells within a clone (*Figure 3F and G*). Results showed that less than 30% of doublets were typically stacked with elevation angles greater than 60° (distal femur, 25%; proximal tibia, 23.2%; *Figure 3F*). In addition, more than half of the doublets oriented orthogonally to the P-D axis (*Figure 3G*). These results are surprising given the observed elongated morphologies of clones. Moreover, perfect rotations, characterized by elevation angles between 80–90°, were rare (5.8% in the DF and 5.6% in the PT; *Figure 3G*).

## Complex longitudinal clones function as columns in the postnatal growth plate

Next, we studied the global orientation of postnatal clones by measuring the angle between the long clone axis and the P-D bone axis (*Figure 4*, *Figure 4—figure supplement 1A and B*). Results revealed the presence of clones with two different morphologies, that is, columns that aligned to the P-D bone axis (PT: 30.5% and DF: 43.9%) and clusters, which oriented orthogonally (PT: 69.4% and DF: 56.1%) (*Figure 4C and G*, *Figure 4—figure supplement 1C*). To assess the possible functions of clusters and columns, we analyzed clone size. We found that zone. Three biologically independent samples were examined in nine independent experiments.

Columns varied in size, ranging from 2 to over 100 cells, many of which traversing the entire length of the growth plate (*Figure 4D and H*). By contrast, most clusters were composed of 2–10 cells and were located directly beneath the secondary ossification center in the RZ and at the very end of the HZ (*Figure 4A, B, E and F*, *Figure 4—figure supplement 1A*). While large columns likely contribute to longitudinal bone growth, the function of small clusters is unclear.

## A column can tolerate 60% incomplete rotations

The main mechanism driving column formation is the rotation of the division plane between sister cells during oriented cell division (*Li et al., 2017*; *Romereim et al., 2014*; *Yuan et al., 2023*). Our findings of two distinct morphologies of postnatal clones and non-stereotypic stacking patterns in embryonic growth plate clones raised the question of the rotational threshold that is required to maintain a columnar structure. To determine the rotation between pairs of cells, we assumed that the final orientation is dictated by the division plane rotation (*Li et al., 2017*; *Romereim et al., 2014*; *Yuan et al., 2023*). Thus, a 90° division plane rotation would result in an elevation angle of 90°, whereas 0° would indicate no rotation (see 'Materials and methods'). As before, we classified elevation angles exceeding 60° as complete rotations, signifying a typically stacked cell doublet oriented along the P-D axis in at least two orthogonal viewing angles (*Figure 1G*). Analysis of hundreds of columns (distal femur, n = 737; proximal tibia, n = 512) revealed that rotations are complete nearly 40% of the divisions (distal femur, 39.5%; proximal tibia, 36.4%) (*Figure 5A and C*). Perfect rotations (80–90°) occurred in less than 10% of the cases (distal femur, 9.6%; proximal tibia, 8.2%) (*Figure 5B and D*). By contrast, clusters exhibited only 15.5% complete rotations in the distal femur and 17.3% in the proximal tibia, with perfect rotations observed in 1.9 and 3% of divisions, respectively.

Lastly, we analyzed the elevation angle as a function of column or cluster size to explore the potential relationship between the two (*Figure 5—figure supplement 1*). Although we did not observe a relationship between cluster size and incomplete rotations, we found that the proportion of the latter increases with column size. Altogether, these results suggest that column formation is resilient, capable of tolerating 60% incomplete rotations. As incomplete rotations accumulate in a clone with every cell division and, subsequently, cross this tolerance threshold, the structure will expand orthogonal to the P-D axis and form a cluster.

## The ratio between growth plate expansion and elongation decreases as the bone grows

Our finding that the majority of embryonic clones formed clusters perpendicular to the P-D axis and that columns appeared only in the postnatal growth plate suggests that a multifunctional design (*Fröhlich et al., 2019*) may allow pre- and postnatal bones to grow differently. Specifically, we

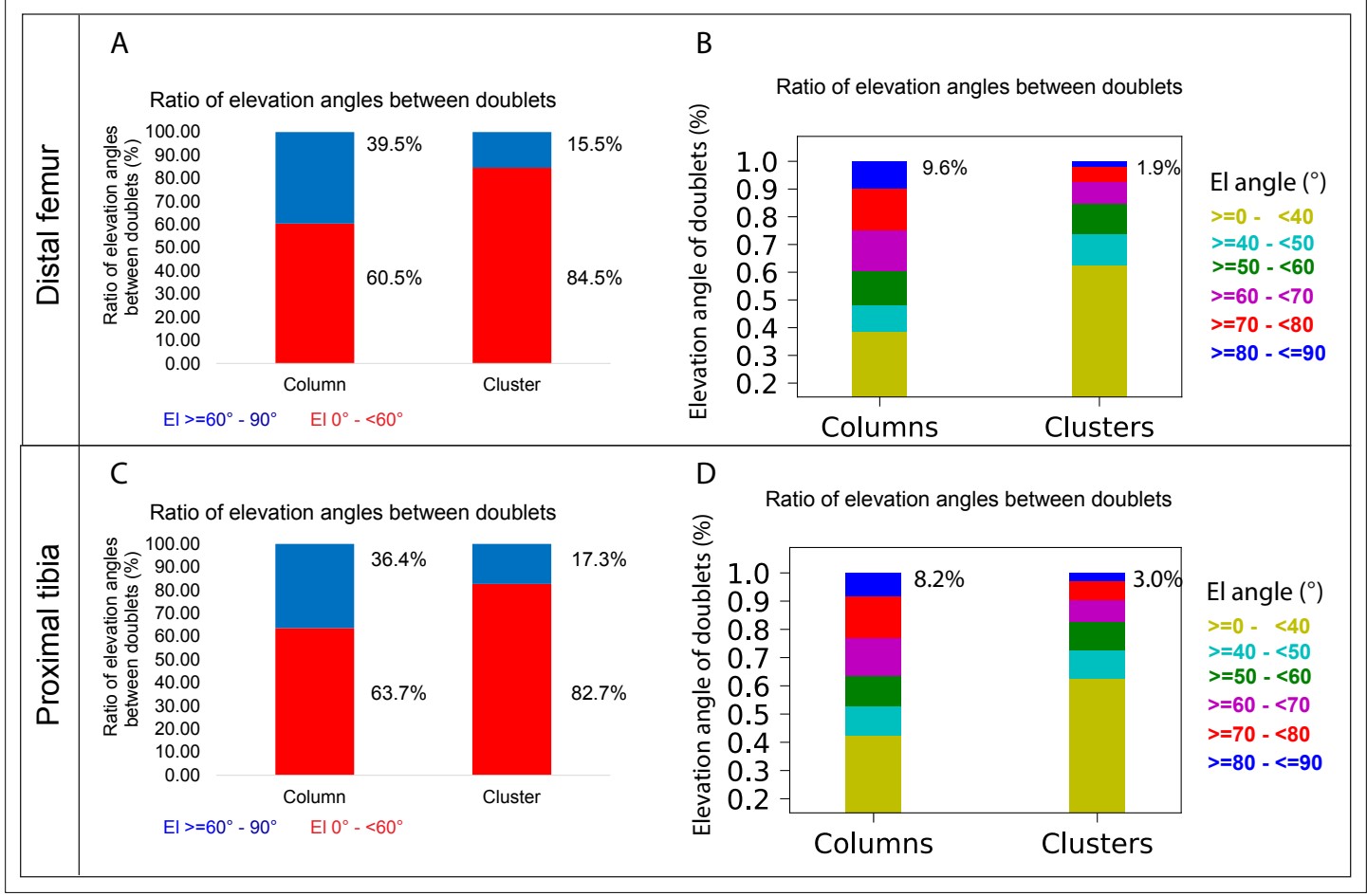

**Figure 5.** A column can tolerate 60% incomplete rotations. (**A, B**) Stacked histograms show the proportion of doublet cells exhibiting complete rotations (i.e., elevation angle [El] of 60–90°, in blue) vs incomplete rotations (El under 60°, in red; **A**) and distribution of elevation angles in columns vs clusters (**B**) in P40 distal femur (DF) growth plates (n = 737 columns, 1129 clusters). (**C, D**) Same analysis in proximal tibia (PT) growth plates (n = 512 columns, 1154 clusters). Elevation angles (°) are color-coded as indicated. but also that bone expansion decreases at a faster rate than bone elongation, resulting in a decrease in the expansion vs elongation (E:E) ratio from embryonic to postnatal stages (*Figure 6B and C*, *Figure 6—figure supplement 1B*). Moreover, E:E growth ratio could be associated with the presence of clusters and columns in the growth plates. For example, in the embryonic growth plate, where columns are rare and clusters abundant, the E:E growth ratio was 0.18 in the DF and 0.16 in the PT. By P40, when columns become abundant and clusters are restricted to periphery, E:E ratio dropped to 0 in the DF and 0.04 in the PT. These trends were observed in all the growth plates analyzed. Notably, in some growth plates, such as the proximal fibula and distal tibia, E:E ratios decreased non-monotonically (*Figure 6C*).

The online version of this article includes the following figure supplement(s) for figure 5:

**Figure supplement 1.** Elevation angles between postnatal doublet cells as a function of clone size.

**Figure supplement 2.** Correlation analysis between cells and nuclei.

**Figure supplement 3.** Noise evaluation of doublet analysis.

hypothesized that while columns support bone elongation, clusters may support bone expansion. This implies that embryonic bones should expand at a higher rate than postnatal bones. To investigate this, we measured the expansion and elongation rates of growth plates from the distal femur, distal and proximal fibula, proximal humerus, distal radius, proximal and distal tibia, and distal ulna at E17.5-E18.5, P14-P16, and P32-P40 (*Figure 6*). The elongation and expansion rates were calculated using registered bones from a previously published database of micro-CT images (*Stern et al., 2015*). Briefly, the elongation rate was calculated as the distance from the median Z-coordinate of the chondro-osseous junction (COJ), representing the end of the growth plate, to the longitudinal origin of the bone, as defined in *Stern et al., 2015*. The expansion rate was calculated as the change in equivalent radius between time points (see 'Materials and methods'; *Figure 6—figure supplement 1*). As shown in *Figure 6A*, isosurface renderings of embryonic and postnatal long bones highlight

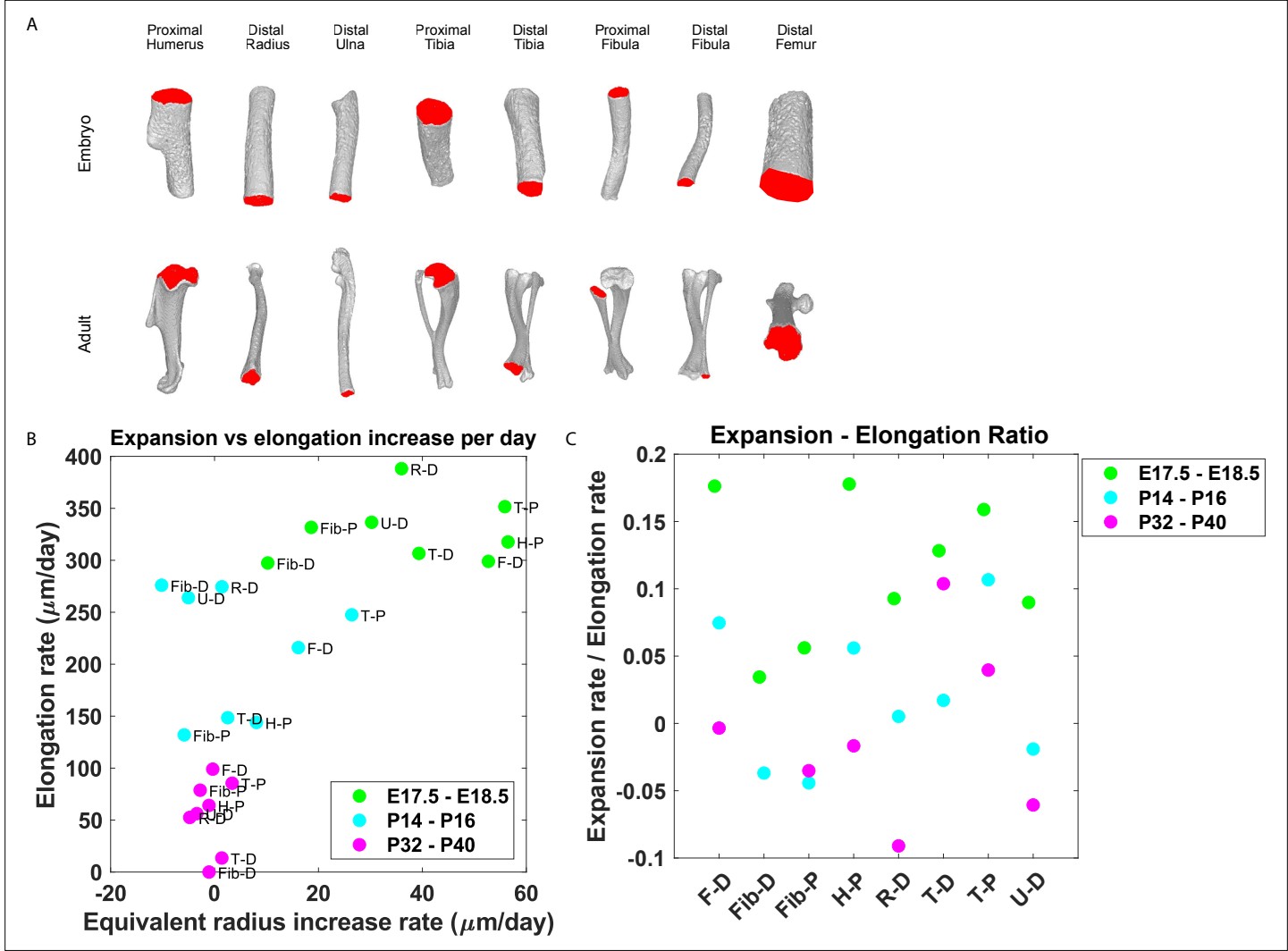

**Figure 6.** Growth plate expansion vs elongation ratio decreases as the bone grows. (**A**) Isosurface renderings of embryonic and postnatal long bones with their chondro-osseous junction (COJ) highlighted in red. Distal femur, distal and proximal fibula, proximal humerus, distal radius, proximal and distal tibia, and distal ulna were analyzed. (**B**) The elongation rate was plotted against the equivalent radius increase rate in micrometers per day for three time windows; E17.5-E18.5 (green), P14-P16 (cyan), and P32-P40 (magenta). Growth plates from each time window clustered together. As bones develop, circumferential expansion is minuscule in comparison to elongation. (**C**) The expansion vs elongation (E:E) ratio, which was plotted for all growth plates and time windows, decreases as bones develop. Interestingly, in the proximal fibula and distal tibia, the decrease was non-monotonic.

The online version of this article includes the following source data and figure supplement(s) for figure 6:

**Source data 1.** Distance from longitudinal origin and equivalent radius per growth plate from E17.5 - P40.

**Figure supplement 1.** Schematic of calculating bone expansion based on micro-CT images and raw data measurements of distance from longitudinal origin and equivalent radius.

the diversity in length and morphology of the different bones and their COJs. In agreement with our hypothesis, we found not only that bones from each stage cluster together.

## Discussion

Chondrocyte columns are a hallmark of growth plate architecture that, in turn, drives bone elongation. In this work, we studied column formation by analyzing the 3D structure of cell clones in the embryonic and postnatal growth plates of mice, using a modified version of 3D MAPs (*Rubin et al., 2021*). Addressing the three criteria for a column, we found that uniclonal and multiclonal columns are rare in the embryonic growth plate. Instead, most clones form elongated clusters that orient orthogonal

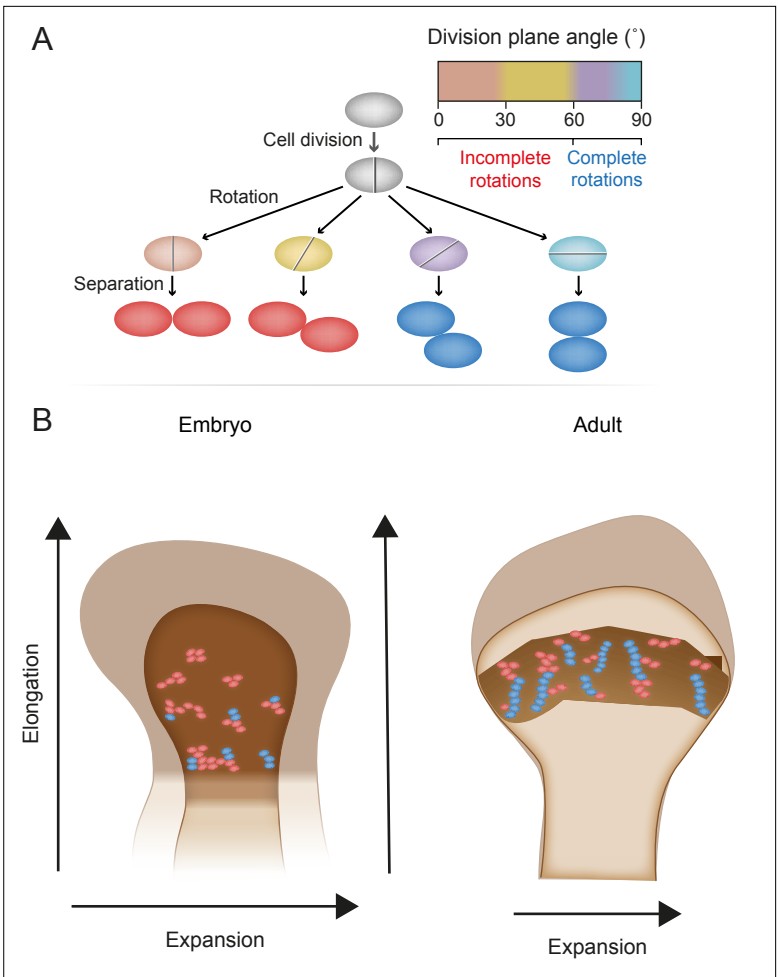

**Figure 7.** Model for the multifunctional design of the growth plate and its relation to bone growth. (**A**) During oriented cell division in the growth plate, the division plane rotation can range from 0° to 90° (brown–blue). Less than 60° rotation indicates an incomplete rotation (red), whereas rotations greater than 60° indicate a complete rotation (dark blue). (**B**) In the embryo, higher rates of incomplete rotations result in cluster formation, which may support the expansion of the growing bone. Postnatally, complete rotations are more frequent, allowing columns to form. This supports bone elongation while potentially limiting bone expansion.

to the longitudinal bone axis as a result of numerous incomplete rotations that occur during cell division within a clone. Postnatally, clones form complex columns from a combination of ordered and disordered stacks of cells, as well as small orthogonally oriented clusters. These morphological entities correlate with the temporal dynamics of growth plate elongation and expansion, suggesting that clusters and columns support different growth strategies during embryonic and postnatal bone development (*Figure 7*).

Column formation is commonly viewed as a key morphogenetic process during bone elongation. During this process, clones of flat PZ cells form stacks, resulting in alignment of the long axis of the column with the longitudinal bone axis (*Dodds, 1930*; *Li and Dudley, 2009*; *Li et al., 2017*; *Romereim et al., 2014*; *Romereim and Dudley, 2011*; *Aszodi et al., 2003*). This arrangement is thought to restrict lateral cell density, thereby maximizing the effect of chondrocyte hypertrophy along the elongation axis of the bone (*Romereim and Dudley, 2011*). Our discovery that during embryogenesis, when bone elongation is at its highest, the growth plates contain only few columns, contradicts previous studies and reveals the need to reconsider the underlying mechanisms. Several possibilities could explain these differences. One plausible explanation is that earlier studies focused on the mechanism of division plane rotation in column formation, but they either analyzed small subsets of clonal doublets without tracking their contributions to columns (*Li et al., 2017*; *Romereim et al., 2014*) or

examined non-clonal doublets (*Yuan et al., 2023*). Without integrating these aspects into a comprehensive study, it is difficult to confirm the existence of clonal columns and determine whether they are formed by cells that have undergone complete division plane rotation. Additionally, it is plausible that the clusters we identified intercalate in a manner that forms geometric non-clonal columns. This could explain the columnar arrangement of chondrocytes observed in histological sections of the growth plate. Moreover, the underlying mechanisms of column formation and the function of bone elongation may be regulated by this organization of clusters into geometric non-clonal columns, or increase in cell volume. The latter has been shown to be a major driver of longitudinal bone growth (*Rubin et al., 2021*; *Breur et al., 1991*; *Wilsman et al., 2008*; *Wilsman et al., 1996*; *Cooper et al., 2013*; *Breur et al., 1997*; *Li et al., 2015*). Changes in chondrocyte morphology, proliferation, and matrix secretion may also play important roles in this process; however, their exact contributions to embryonic bone elongation are unclear.

As the cellular entity that regulates bone growth and morphology, the growth plate must balance between different requirements and constraints while performing its functions. It is therefore reasonable to assume that the growth plate follows the 'multifunctional design' principle. This concept addresses the need to reduce the size and complexity of systems which perform multiple tasks by implementing several functions within a single structure (*Fröhlich et al., 2019*). Our finding that embryonic bones extensively expand as they elongate supports the existence of a multifunctional design mechanism that allows coping with these different growth requirements. Elongated clusters that orient orthogonal to the longitudinal axis due to incomplete rotations during cell divisions might be the cellular strategy that enables growth plate expansion in the embryo. In line with this idea, disruption of cell polarity, which leads to more incomplete division plane rotations, causes bone widening (*Rubin et al., 2021*; *Ahrens et al., 2009*; *Gao et al., 2011*; *Aszodi et al., 2003*; *Kuss et al., 2014*), whereas recent simulation of the growth plate predicted that lack of rotation during cell division causes the bone to expand in width (*Yokoyama et al., 2024*). Later, during postnatal stages, bone expansion decreases dramatically while longitudinal growth is maintained. Correlating with this change in growth strategy is the appearance of columns in the center of the growth plate and small orthogonal clusters on the edges. Interestingly, these small clusters were observed nearly a century ago by *Dodds, 1930* in the RZ of human phalanx growth plates. Our current analysis of these different morphologies suggests that columns contribute to bone elongation, as previously thought, whereas clusters may provide a mechanism to control local morphology of the growing bones. Additionally, clones in articular cartilage were shown to form by non-stereotypic cell stacking, similar to the clusters we observed, raising the question of the function they may serve (*Decker et al., 2017*).

The rotation of the division plane is the cellular mechanism that underlies the formation of columns versus clusters. If indeed these cellular modalities are utilized to promote different growth strategies, then the mechanism that controls the division plane can be central in the regulation of bone morphology. Moreover, it can provide an adaptable means to shift between different growth strategies such as elongation, expansion, and curvature formation. Interestingly, a simulation of cell adhesion in the growth plate showed that considering the mechanical confinement of the chondrocyte matrix, a complete rotation is energetically favorable (*Yokoyama et al., 2024*). Thus, the low rates of complete division plane rotations, deduced from the doublet elevation angle analysis, suggest that the orientation of the division plane is tightly regulated. While the mechanisms that regulates the rotation of the division plane during bone growth are not fully understood, previous work provides several indications for both mechanical and molecular signals. These include Fz/Vangl/PCP signaling, integrins and muscle force, as well as matrix composition (*Li and Dudley, 2009*; *Li et al., 2017*; *Ahrens et al., 2009*; *Shwartz et al., 2012*; *Gao et al., 2011*; *Yang et al., 2003*; *Yuan et al., 2023*; *Aszodi et al., 2003*; *Bengtsson et al., 2005*; *Yang and Mlodzik, 2015*; *Killion et al., 2017*; *Pierantoni et al., 2021*; *Kuss et al., 2014*; *Prein et al., 2016*). It will be interesting to revisit the role of these factors and pathways in the embryonic growth plate in light of our new findings.

The observation of chondrocyte columns in P40 growth plates raises the question of when columns start to form. Previous studies showed a switch in column clonality when the secondary ossification center forms (*Newton et al., 2019*; *Mizuhashi et al., 2018*; *Hallett et al., 2022*). Therefore, it is reasonable to speculate that clones switch from orthogonally oriented clusters to parallel columns at that time point.

One limitation of our rotation analysis is that in order to calculate the relative position between neighboring cells, we assumed that chondrocyte rotation and orientation are correlated and that chondrocytes do not move after rotating, as was previously suggested (*Romereim et al., 2014*; *Yuan et al., 2023*). Moreover. growth plates of various organisms employ different strategies to create bones of diverse morphologies, which serve different functions such as running, swimming, flying, etc. Thus, it is possible that in each context, the ratio between clusters and columns is adjusted to cope with these different requirements.

Here, we discovered that the core mechanism underlying chondrocyte column formation, namely a complete rotation of the division plane during chondrocyte proliferation, is rare in the embryonic growth plate. Our 3D analyses reveal the temporal dynamics of division plane rotations and their effect on the formation of columns and clusters in the pre- and postnatal growth plate, as well as the correlation of these morphogenetic processes with different growth behaviors. Overall, these findings establish a new model for column formation and provide a new understanding of the cellular mechanisms underlying growth plate activity and bone elongation and expansion during development.

# Materials and methods

**Key resources table**

| Reagent type (species) or resource | Designation | Source or reference | Identifiers | Additional information |
|---|---|---|---|---|
| Strain, strain background (mouse) | Col2-CreER$^T$ | Jackson Laboratories | RRID:IMSR_JAX:006774 | |
| Strain, strain background (mouse) | R26R-Confetti | Jackson Laboratories | RRID:IMSR_JAX:017492 | |
| Sequence-based reagent | Col2CreER-F | Jackson Laboratories | PCR primers (stock # 006774) | CAC TGC GGG CTC TAC TTC AT |
| Sequence-based reagent | Col2CreER-R | Jackson Laboratories | PCR primers (stock # 006774) | ACC AGC AGC ACT TTT GGA AG |
| Sequence-based reagent | Confetti-mutant Forward | Jackson Laboratories | PCR primers (stock # 017492) | GAA TTA ATT CCG GTA TAA CTT CG |
| Sequence-based reagent | Confetti-WT Forward | Jackson Laboratories | PCR primers (stock # 017492) | AAA GTC GCT CTG AGT TGT TAT |
| Sequence-based reagent | Confetti-common | Jackson Laboratories | PCR primers (stock # 017492) | CCA GAT GAC TAC CTA TCC TC |
| Chemical compound, drug | Tamoxifen | Sigma-Aldrich | T-5648 | |
| Other | Draq5 | Thermo Scientific | 62252 | 1:2000 for embryonic samples and 1:1500 for postnatal samples |

## Animals

For genetic labeling of chondrocyte clones in embryonic and postnatal growth plates, Col2a1-CreER:R26R-Confetti heterozygous mice were generated by crossing mice homozygous for Col2-CreER$^T$ (Jackson Laboratories, FVB-Tg (Col2a1-cre/ERT)KA3Smac/J; *Nakamura et al., 2006*) with R26R-Confetti homozygotes (Jackson Laboratories, B6.129P2-*Gt(ROSA)26Sor*$^{tm1(CAG-Brainbow2.1)Cle}$/J; *Snippert et al., 2010*). Mice were dissected in cold phosphate-buffered saline (PBS), fixed for 3 hr at 4°C in 4% paraformaldehyde (PFA), washed in PBS, and stored at 4°C in 0.5 M EDTA (pH 8.0, Avantor Performance Materials) with 0.01% sodium azide (Sigma) for 2 days. Limb samples were then dehydrated in 30% sucrose/PBS overnight at 4°C, embedded in OCT, and stored at –80°C the following day. In all timed pregnancies, the plug date was defined as E0.5. For harvesting of embryos, timed-pregnant female mice were sacrificed by CO$_2$ exposure. Embryos were sacrificed by decapitation with surgical scissors, and postnatal mice were sacrificed by CO$_2$ exposure. Tail genomic DNA was used for genotyping by PCR (Key Resources Table). All animal experiments were pre-approved by and conducted according to the guidelines of the Institutional Animal Care and Use Committee (IACUC) of the Weizmann Institute (IACUC 01750221-1 and IACUC 05700723-2). All animals used in this study had access to food and water ad libitum and were maintained under controlled humidity and temperature (45–65%, 22 ± 2°C, respectively). For each experiment, three mice were collected from

at least two independent litters. Mouse embryos were used regardless of their sex, whereas postnatal experiments were performed only on females to control for potential sex-related phenotypes.

For clonal genetic tracing, tamoxifen was administered by oral gavage (Fine Science Tools) at a dose of 3 mg to P30 *Col2a1-CreER:R26R-Confetti*+/- mice or 2 mg to time-mated *R26R-Confetti* females at E14.5. Tamoxifen (Sigma-Aldrich, T-5648) was dissolved in corn oil (Sigma-Aldrich, C-8267) at a final concentration of 15 mg/ml. Neighboring cells that expressed the same fluorescent protein were considered clonal.

## Sample preparation

200-µm-thick sagittal cryosections of the embryonic or postnatal right hindlimbs from *Col2a1-CreER:R26R-Confetti*+/- mice were collected into a 12-well plate filled with 1 ml PBS. To remove OCT, samples were washed twice with PBS at room temperature (RT) with gentle rocking. Then, nuclei were stained with Draq5 (Thermo Scientific 62252) diluted in PBST (PBS + 0.1% Triton X-100) for 2 hr at RT at a dilution of 1:2000 for embryonic samples and 1:1500 for postnatal samples. Three sections from the central region of the proximal tibia and distal femur growth plates were selected for further processing, together covering 600 µm along the medial-lateral bone axis. Sections were then placed in Rims (*Rubin et al., 2021*) with a refractive index of 1.45 (74% Histodenz/PB) overnight at RT, and then mounted the following day with Rims between a glass slide and coverslip. Because Confetti fluorophores fade quickly, sections were imaged within 1 week of preparation. A 200-µm-thick section contains 5–6 cells in thickness in the PZ and 2–3 cells in the HZ.

## Image acquisition

The proximal tibia and distal femur growth plates were imaged by a combination of multiphoton and confocal imaging using an upright Leica TCS SP8 confocal laser-scanning/MP microscope (Leica Microsystems, Mannheim, Germany), equipped with external non-descanned detectors (NDD) HyD and HyD SP GaAsP detectors for confocal imaging (Leica Microsystems CMS GmbH, Germany). Channels were collected in sequential mode. The CFP signal from the Confetti was excited by 900 nm laser line of a tunable femtosecond laser 680–1080 Coherent vision II (Coherent GmbH, USA). Emission signal was collected using an external NDD HyD detector through a long pass filter of 440 nm. The GFP and YFP Confetti signal was excited by an Argon laser and collected with HyD SP GaAsP internal detectors (Ex 488 nm Em 498–510 nm and Ex 514 nm Em 521–560 nm). The RFP Confetti signal was excited by a DPSS 561 nm laser with emission collection at 582–631 nm and the Draq5 signal was excited by a HeNe 633 laser, with emission collection at 669–800 nm. As reported previously (*Newton et al., 2019*), we rarely observed GFP clones in the growth plate.

Growth plates were imaged as a z-stack using a galvo scanner through a HC PL APO ×20/0.75 CS2 objective (scan speed, 400 Hz; zoom, 0.75; line average, 4; bit depth, 8; Z step, 0.39 µm). For embryonic samples, a format of 4096 × 4096 (XY) was used resulting in a pixel size of 180 nm (XY) and for postnatal samples, a format of 2000 × 2000 (XY) produced a pixel size of 369 nm (XY). Z stacks were acquired using the galvo scanner (objective movement) at 0.39 µm intervals.

Embryonic samples were imaged with a single field of view (FOV), which covered the middle of the RZ through the beginning of the COJ. Postnatal samples were imaged with multiple overlapping FOVs (10% overlap), which covered the entire growth plate from the bottom of the secondary ossification center through the beginning of the COJ. Postnatal images were stitched in ImarisStitcher 9.9.0 (Bitplane).

## Growth plate segmentation

To generate a mask of the growth plate, Microscopy Image Browser (version 2.81) (*Belevich et al., 2016*) was used to manually segment the region between the secondary ossification center and the COJ in postnatal images and the entire growth plate region until the COJ in embryonic images. Additionally, a mask of the HZ was created by identifying the cells with the stereotypic chromatin staining unique to this zone.

## Nuclei segmentation

Images of fluorescently stained nuclei were automatically segmented as described previously (*Rubin et al., 2021*; *Bartschat et al., 2016*). For embryonic images, a Gaussian blur filter (radius 2) and

background subtraction (rolling ball radius 25) was applied in Fiji (*Schindelin et al., 2012*) prior to segmentation. We used standard deviations of $\sigma = 12$ for RZ, PZ, and PHZ nuclei and of $\sigma = 25$ for HZ nuclei for embryonic images and $\sigma = 5$ for RZ, PZ, and PHZ nuclei and of $\sigma = 8$ for HZ nuclei for post-natal images. Subsequently, local intensity maxima were extracted from the LoG-filtered image and reported as potential nuclear centers. For each potential seed point, we computed the mean intensity in a $4 \times 4 \times 4$ voxel-wide cube for embryonic samples and $2 \times 2 \times 2$ voxel-wide cube for postnatal samples surrounding the centroid. All image analyses were performed on a Windows Server 2012 R2 64-bit workstation with 2 Intel(R) Xeon(R)CPU E5-2687W v4 processors, 512 GB RAM, 24 cores and 48 logical processors.

## Clone segmentation

To generate a mask of each clone, .lif files were converted to.ims files using the Imaris file converter (version 9.8.0). Imaris surfaces tool was used to create surfaces and extract the three clone masks (yfp, rfp, and cfp) from the image; Gfp clones were not present in the images. Surfaces were created with a grain size of 1.00 µm using the absolute intensity feature, with thresholds varying depending on the image. Surfaces with volumes greater than 150 µm³ were kept for further analysis. Next, the clone masks were overlapped with the raw signal in Fiji to inspect the quality of the segmentation. Quality was high in images where clones did not touch each other. In images with high labeling efficiency, where clones touched each other, some clone segmentations needed to be manually corrected in MIB.

## Nucleus and clone feature extraction

Following image segmentation and masking, separate segmented images were created for each clone mask in Fiji (S1 data) by assigning pixels outside of the clone mask and growth plate mask a value of 0. Next, segmented clone and nuclear images were relabeled using Morpholib plugin (*Legland et al., 2016*) and converted to 16- or 32-bit float depending on the number of objects in each image (S2 data). Images were reoriented in Fiji (*Schindelin et al., 2012*) so that the proximal-distal bone axis aligned to the Y-axis of the image coordinate. Nucleus and clone features were extracted as described previously (*Rubin et al., 2021*) in MATLAB (version R2017b) with a volume range of 100–1200 µm for nuclei and 150–500 × 10⁷ µm for clones.

## Morphometric analysis

Clone morphometrics, such as PC coefficients and PC orientations, were calculated as described previously (*Rubin et al., 2021*). Cluster, column, or clone size was defined as the number of nuclei per cluster, column, or clone. These morphometric features were displayed as a histogram, a 3D morphology map, or both.

## Correlation analysis between cells and nuclei

To evaluate whether nuclear centroids can be used as a proxy for cell centroids, we performed correlation analysis on our previously published dataset of a wild-type tibia sample (*Rubin et al., 2021*). This dataset included segmentation data on both nuclei and cells from the entire growth plate, but not clone information. We matched between cells and nuclei in the dataset by calculating pairwise distances between centroids of cells and nuclei. Pairs were considered matched if the centroid distance was less than 20 µm and the 'closest cell to a nucleus' was a mutual neighbor to the 'closest nucleus to a cell', resulting in 45,166 paired cells and nuclei.

To simulate random clones in the tibia sample, we generated clones of various sizes, radii, and random positions along the P-D axis. These parameters were sampled from real clone sizes in the embryonic distal femur sample. Within the pool of matched cell–nucleus pairs, we determined the number of cell centroid positions falling within the randomly sampled sphere radius and P-D positions, allowing a variation of up to 15% of the P-D height on either side. While cells meeting these criteria may or may not match the desired clone size, we calculated pairwise distances for these cells, sorting them from largest to smallest. Using the single-connected component method, detailed in the 'Doublet quantification' subsection, we removed the farthest-apart cells and continued this process to obtain single connected components of cells with the desired clone size.

A total of 1278 random clones were generated in the paired data. Within these clones, we calculated the elevation angle for cell doublets and nuclear doublets separately and visualized the variation between them in a histogram (*Figure 5—figure supplement 2A*). The results indicated a striking similarity in histogram distribution. Additionally, we computed the Pearson correlation between the mean elevation angle in a clone based on either cell or nuclear centroids, resulting in a correlation value of 0.79 (*Figure 5—figure supplement 2B*). These results suggest that elevation angle calculation performed on nuclei can serve as a reliable proxy for cell-based measurements.

## Doublet quantification

For doublet quantification in clones, the process starts by measuring distances between the centroids of all possible pairs of nuclei to identify the nearest neighbor pairs. These distances are sorted from low to high, establishing them as edges in the graph. For instance, in a clone with five nodes, there are 10 potential edges. The single-connected component method is applied to understand the graph's topology and the distribution of nuclei within the clone. In a linear or columnar topology, a connected component should have a maximum of four edges (e.g., 1–2, 2–3, 3–4, 4–5). In a spherical-like topology, a connected component can have all 10 edges. The analysis begins with the smallest edge in the list. It is checked if this edge forms a single connected component. If not, the next smallest edge is added in the list, and this process continues. When a single connected component is achieved, the addition of edges stops. The final list of edges obtained in this manner defines the clone's topology, with each edge termed as a doublet. These edges, present in the final list, are utilized as total number of doublets for the subsequent elevation angle analysis.

To evaluate the potential noise introduced into the measurement by measuring all nuclei pairs, the proportion of elevation angles as a function of number of nuclei neighbors per columns and clusters was calculated as well as the population statistics (*Figure 5—figure supplement 3*). We observed consistent patterns for both columns and clusters at any chosen number of nuclei neighbors. The range of the proportion of elevation angles between 60 and 90° for 1–5 neighbors in columns was 13.2–15.5% for the distal femur and 16.3–21% for the proximal tibia. For clusters, the range was 27.8–35.5% for the distal femur and 29.1–36.5% for the proximal tibia. The mean elevation angle distribution (60–90°) from 1 to 10 neighbors is 14.5% (columns) and 28.4% (clusters) in the distal femur and 19.5% (columns) and 29.7% (clusters) in the proximal tibia (*Figure 5—figure supplement 3A–D*). The proportion of elevation angles up to 10 nuclei neighbors within columns and clusters represents up to 99.1% (distal femur: 99.1%; proximal tibia: 98.8%) of all nuclei doublet pairs in columns and up to 71.6% (distal femur: 71.6%; proximal tibia: 62%) in clusters (*Figure 5—figure supplement 3E and F*). While examination of the influence of increasing neighbors on the proportion of elevation angles between 60 and 90° did not show any statistical significance in columns, we found clusters to show statistical significance (p-value<0.05, two-sample *t*-test) when comparing the mean of lower number of neighbors (*Lecuit and Lenne, 2007*; *Irvine and Wieschaus, 1994*; *Bailles et al., 2022*; *Collinet and Lecuit, 2021*; *Sutherland et al., 2020*) with larger numbers of nuclei neighbors (*Sutherland et al., 2020*; *Rubin et al., 2021*; *Breur et al., 1991*; *Wilsman et al., 2008*; *Wilsman et al., 1996*) for both the distal femur and proximal tibia (*Figure 5—figure supplement 3C and D*). This suggests that measuring elevation angle of nuclei doublets in clusters has slight noise present due to the spatial organization of nuclei in the clone, while the variation in columns is insignificant.

## Calculation of elevation angle and the angle between clone and PD axis

To determine the elevation angle between nuclei in a doublet, we first transformed the Cartesian coordinates of doublet to a spherical coordinate system, in which the Z-axis represents the P-D bone axis. Then, we shifted the mean position of a doublet to the origin zero and used the MATLAB function cart2sph to obtain the elevation angle (phi) and the radius (r). The elevation angle was measured as the angle between the projected line of a nucleus on the XY plane to the line connecting two nuclear centroids (*Figure 1F*). The XY plane is perpendicular to the Z-axis. If one nucleus in the doublet has an elevation angle of phi, the angle of the other nucleus is -phi. The elevation angle values are in the range of [-pi/2, pi/2]. The radius in the spherical coordinate system is equivalent to half of the distance between two nuclei in a Cartesian coordinate system. The elevation angle between two nuclei lying on top of one another is 90°, whereas the angle between two nuclei that are positioned next to each

other in the XY plane is 0. Because the elevation angles of nuclei in a doublet mirror each other, we only used the absolute value.

To determine the distribution of elevation angles, we divided them into seven categories: between 0 and <15° (magenta), between 15 and <30° (yellow), between 30 and <45° (cyan), between 45 and <60° (green), between 60 and <70° (orange), between 70 and <80° (red), and between 80 and 90° (blue). The mean value of each category is reported as the elevation angle for column or cluster clone.

The angle between clone and PD axis (theta) was calculated using the MATLAB function 'theta = atan2(norm(cross(u,v)),dot(u,v))', where u and v are two vector representing PD axis and clone orientation. If the function value exceeded 90°, the reported value was (180°-theta). Finally, to use the same notation as for the elevation angle analysis, the PD-PC angle is reported as (90°-theta). Thus, if the longest axis of the clone (PC1) is parallel to PD axis, then the reported value is 90°, and if it is perpendicular, then the reported value is 0°.

## Multiclone formation analysis through clone merging

In our analysis of embryonic data, clones were merged based on the criterion that the distance between any nucleus within one clone to any neighboring clone was less than 15 µm. After the merging process, the total number of nuclei in a merged clone equaled the sum of nuclei in the original individual clones. All properties of merged clones were computed in the same manner as for individual clones.

## Quantification of growth plate elongation and expansion rates

### Micro-CT image dataset

We used a previously published micro-CT database (*Stern et al., 2015*) to analyze eight different growth plates at three developmental stages (E17.5-E18.5, P14-P16, and P32-P40). The bones of C57/Bl6 mice were scanned using a high-resolution eXplore Locus SP micro-CT scanner for embryonic stages and a TomoScope 30S Duo scanner for postnatal stages, ensuring isotropic resolutions of 7.139 µm$^3$ and 36 µm$^3$, respectively. We removed low-quality images yielding four to eight bones per embryonic growth plate and two to four per postnatal growth plate.

### Data preparation

We ensured anatomical correspondence across time points using a bone image registration algorithm (*Stern et al., 2015*).The *Autocontext* image segmentation module from *Ilastik* (*Berg et al., 2019*) facilitated the segmentation of mineralized bone, and manual removal of secondary ossification at postnatal stages was conducted using *Fiji* (*Schindelin et al., 2012*).

### Extraction of COJ

Post-registration bones were aligned such that their proximal-distal axis was vertical. We generated images where the value of each bone voxel was its Z-coordinate, enabling identification of the COJ through a maximum projection that highlighted the highest Z-values. Manual thresholding and corrections isolated the COJ, with an additional mirroring step for distal growth plates. Raw data is available in *Figure 6—source data 1*.

### Calculation of growth and equivalent radius

To quantify elongation, we extracted the median Z-coordinate of the COJ voxels and calculated the distance to the longitudinal origin of the bone, defined as the thinnest point of the cartilaginous template from previous research (*Stern et al., 2015*).We then calibrated voxel units to physical units for accuracy. For the equivalent radius, we applied principal component analysis (PCA) to the COJ voxel coordinates to align the surface with the XY plane, calculating the area of the projected voxels on this plane. The equivalent radius was derived from the area using $r = \sqrt{A/\pi}$, where $r$ is the radius and $A$ is the area, with values calibrated to physical units. Raw data is available in *Figure 6—source data 1*.

## Code availability

The codes utilized in the current study are available on Github at the following link: https://github.com/ankitbioinfo/clonal_analysis_in_growth_plates_elife; (copy archived at *Agrawal, 2024*).

## Acknowledgements

We thank Nitzan Konstantin for editorial assistance and members of the Zelzer lab for their advice and encouragement throughout this project. We thank Phillip T Newton and Andrei S Chagin for sharing Confetti labeled images for us to compare our data to. We thank M Vijay Kumar, Uri Alon, Efi Efrati, Ariel Amir, Hillel Aharoni, and Jure Leskovec for useful discussions. We thank the de Picciotto Cancer Cell Observatory in memory of Wolfgang and Ruth Lesser, Weizmann Institute of Science, for providing confocal/multiphoton imaging infrastructure, Ishai Sher and Hanna Vega from the Graphic Design Department at the Weizmann Institute of Science for their help with graphics. This study was supported by grants from Israel Science Foundation (ISF) Breakthrough Research Grants ('MAPATZ') 1387/23, Weizmann-Sagol Institute for Longevity Research, and the Julie and Eric Borman Family Research Funds (EZ). TS and OG were supported by the University of Michigan School of Dentistry startup funds, and MJL received support from the Oral Health Sciences PhD program.

## Additional information

### Funding

| Funder | Grant reference number | Author |
| --- | --- | --- |
| Israel Science Foundation | 1387/23 | Elazar Zelzer |
| Weizmann - Sagol Institute for Longevity Research | | Elazar Zelzer |
| Julie and Eric Borman Family Research Funds | | Elazar Zelzer |
| University of Michigan School of Dentistry | Startup funds | Tomer Stern |
| University of Michigan Oral Health Sciences PhD program | | Meng-Jia Lian |

The funders had no role in study design, data collection and interpretation, or the decision to submit the work for publication.

### Author contributions

Sarah Rubin, Conceptualization, Data curation, Formal analysis, Investigation, Visualization, Methodology, Writing - original draft, Writing - review and editing; Ankit Agrawal, Conceptualization, Formal analysis, Visualization, Methodology, Writing - review and editing; Anne Seewald, Data curation, Formal analysis, Writing - review and editing; Meng-Jia Lian, Olivia Gottdenker, Formal analysis; Paul Villoutreix, Adrian Baule, Conceptualization, Writing - review and editing; Tomer Stern, Conceptualization, Formal analysis, Writing - review and editing; Elazar Zelzer, Conceptualization, Supervision, Funding acquisition, Writing - original draft, Project administration, Writing - review and editing

### Author ORCIDs

Sarah Rubin (ID) https://orcid.org/0000-0003-0601-8802
Ankit Agrawal (ID) https://orcid.org/0009-0006-1700-2397
Anne Seewald (ID) https://orcid.org/0000-0002-4904-2063
Paul Villoutreix (ID) https://orcid.org/0000-0002-6333-5735
Elazar Zelzer (ID) https://orcid.org/0000-0002-1584-6602

### Ethics

All animal experiments were pre-approved by and conducted according to the guidelines of the Institutional Animal Care and Use Committee (IACUC) of the Weizmann Institute (IACUC 01750221-1 and IACUC 05700723-2). All animals used in this study had access to food and water ad libitum and were maintained under controlled humidity and temperature (45-65%, 22 ± 2 °C, respectively).

### Decision letter and Author response

Decision letter https://doi.org/10.7554/eLife.95289.sa1

Author response https://doi.org/10.7554/eLife.95289.sa2

## Additional files

### Supplementary files
- Supplementary file 1. Mean and std for bone elongation and expansion measurements.
- MDAR checklist

### Data availability
The datasets generated and analyzed during the current study are available on Zenodo at the following links: https://doi.org/10.5281/zenodo.10440013, https://doi.org/10.5281/zenodo.10444731, https://doi.org/10.5281/zenodo.10446055, https://doi.org/10.5281/zenodo.10446092, https://doi.org/10.5281/zenodo.10446121, https://doi.org/10.5281/zenodo.10446131, https://doi.org/10.5281/zenodo.10446123, https://doi.org/10.5281/zenodo.10446145.

The following datasets were generated:

| Author(s) | Year | Dataset title | Dataset URL | Database and Identifier |
|---|---|---|---|---|
| Rubin S | 2023 | Col2creER;Confettihet_gE14.5_E18.5_litter154_m3 | https://doi.org/10.5281/zenodo.10440013 | Zenodo, 10.5281/zenodo.10440013 |
| Rubin S | 2023 | Col2creER;Confettihet_gE14.5_E18.5_litter154_m4 | https://doi.org/10.5281/zenodo.10444731 | Zenodo, 10.5281/zenodo.10444731 |
| Rubin S | 2023 | Col2creER;Confettihet_gE14.5_E18.5_litter153_m7 | https://doi.org/10.5281/zenodo.10446055 | Zenodo, 10.5281/zenodo.10446055 |
| Rubin S | 2023 | Col2creER;Confettihet_gP30_P40_litter152_m3_DF | https://doi.org/10.5281/zenodo.10446092 | Zenodo, 10.5281/zenodo.10446092 |
| Rubin S | 2023 | Col2creER;Confettihet_gP30_P40_litter152_m3_PT | https://doi.org/10.5281/zenodo.10446121 | Zenodo, 10.5281/zenodo.10446121 |
| Rubin S | 2023 | Col2creER;Confettihet_gP30_P40_litter152_m4_DF | https://doi.org/10.5281/zenodo.10446131 | Zenodo, 10.5281/zenodo.10446131 |
| Rubin S | 2023 | Col2creER;Confettihet_gP30_P40_litter151_m2 | https://doi.org/10.5281/zenodo.10446123 | Zenodo, 10.5281/zenodo.10446123 |
| Rubin S | 2023 | Col2creER;Confettihet_gP30_P40_litter152_m4_PT | https://doi.org/10.5281/zenodo.10446145 | Zenodo, 10.5281/zenodo.10446145 |

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
