## [Editor Report]

The study presents a landmark finding on quantifying the orientation and organization of chondrocyte columns in the prenatal and postnatal growth plate cartilage using advanced 3D imaging and a sophisticated image analysis pipeline. The evidence supporting the authors' conclusions regarding clusters of chondrocytes (instead of columns) in the fetal growth plate is considered compelling, with rigorous imaging analyses. The work will be of broad interest to developmental and cell biologists.

---

## [Decision Letter]

**Decision letter after peer review:**

Thank you for submitting your article "Bone elongation in the embryo occurs without column formation in the growth plate" for consideration by *eLife*. Your article has been reviewed by 3 peer reviewers, and the evaluation has been overseen by a Reviewing Editor and Kathryn Cheah as the Senior Editor.

Essential revisions (for the authors):

1) Prove the validity of the methodology that the authors used to measure nuclear elevation angles and define clonal orientations (morphometric analyses: are fetal clone sizes sufficiently large to reflect the orientation?) – related to comments by Reviewer 2 and 3.

2) Establish the correlation between column/cluster organization and length-width ratios by calcein dye front labeling (or another method) to measure the change in the growth cartilage diameter over the same time – related to comments by Reviewer 1 and 2.

3) Stratify clone data according to the position in the resting and proliferative zones (top, middle, and bottom of each zone) to map clone development over time – related to comments by Reviewer 2.

4) Address inconsistencies with the literature and provide a more balanced consideration of the past findings to place the current findings better in context. Also, acknowledge the limitations of the prior and current analyses, and dampen/soften the conclusions/statements as appropriate – related to comments by all reviewers.

*Reviewer #1 (Recommendations for the authors):*

With the shape data you already have in your group, you might be able to quantify the rates of longitudinal vs circumferential growth in the region studied in the present work. You would then be able to test the hypothesis that the change in orientation of the columns is correlated (or not) with the dominant direction(s) of growth over the time period of interest.

It would be nice to make the code for the analysis pipeline available publicly in the interest of open science.

*Reviewer #2 (Recommendations for the authors):*

Overall, this is an interesting study that introduces a more sophisticated clonal generation method and presents a new approach to data collection and analysis that together may also have high potential more generally for studies of tissue growth and morphology. However, there are several major issues that need addressing to increase confidence in these results and conclusions:

1. The title "Bone elongation in the embryo occurs without column formation in the growth plate" is misleading for two reasons: (1) Figure 1 clearly shows columns of cells in the embryo – as acknowledged by the authors – and (2) the conclusion results from correlative studies and was not tested by experimental manipulation. One should not make such strong statements in the absence of mechanistic data or the presence of conflicting evidence. State what the data show, not what you want it to show.

2. The main argument in this manuscript appears to be more semantic than substantive. The authors report that the vast majority of clones in the embryo exist as clusters, yet 2D imaging of the embryo mostly reveals short columns (as stated above). The authors state that the cluster arrangement of clones can only be visualized in 3D. This suggests that "clusters" are possibly "clusters of columns" and not clusters of the structure diagrammed in Figure 6. As mentioned in the first paragraph, we already knew that not all cells rotate completely (some partial and some not) after division and therefore a single clone forming adjacent columns or "branching" is expected at some frequency based on the efficiency and geometry of rotation. It might be more accurate to state the current work shows that the efficiency of rotation is less or that under-rotating cells have a greater effect on clone morphology than previously thought. Either way, this is a refinement, not a violation, of the current mechanistic paradigm.

3. The emphasis on nuclear elevation measurements is not particularly convincing. Whether the alignment of nuclei reflects rotation or is simply the most efficient packing method for two cells in a constrained environment is not certain. Presenting the expectation of nuclear stacking is a "straw man" argument.

4. Concerns about the measured data extend to the methodology. In general, the final "calculated" model should reflect what is observed in the raw data. The "clusters" described in the graphs or schematized in Figure 6 are not visible in the raw data. The authors mention the 2D vs. 3D problem, but this is not the issue. A single section (physical or optical) should cut through different clusters at different angles. The final result should be an image containing clone morphologies at the ratio in which they appear in the tissue. This is especially true for clones arranged medial to lateral – the predominant clone type reported – given the sagittal sections and radial symmetry of the growth plate. Unfortunately, the graphical data and the raw image data appear in conflict and therefore I do not have high confidence in this analysis and the resulting model.

5. Although clonal analysis is a powerful technique, this specific study is predicated on the assumption that the clones observed in the proliferative zone originated from recombination events in single proliferative zone cells. However, a confounding factor is that endochondral ossification progressively removes proliferative chondrocytes through hypertrophy and remodeling of the calcified cartilage matrix. This means that the short (P-D axis) clones in the short proliferative zone of embryonic growth plates have limited resident lifetimes, which based on the growth rate is probably 24-48 hours. This would not affect the analysis of the longer clones of the postnatal growth plate that would require substantially longer periods to be completely removed. This is an important distinction because the authors induce recombination (cell labeling) in the embryo 96 hours before performing the analysis. Thus, most of the clones that were analyzed are likely to have derived from recombination events that occurred in the resting zone. Previous work demonstrated that resting zone clones expand in clusters and laterally rather than in columns as is common with proliferative chondrocytes. Therefore, a resting zone clone that enters the proliferative zone could appear as a single column (if labeled after the last resting cell division), or as a cluster of columns or a mixture of columns and other cell arrangements depending on how many cells in the resting clone are synchronously incorporated into the proliferative zone and how many cells in the new proliferative cluster divide before analysis. This is potentially a fatal flaw for this study.

*Reviewer #3 (Recommendations for the authors):*

Review attachment:

https://submit.elifesciences.org/*eLife*_files/2024/01/02/00127070/00/127070_0_attach_6_4060_convrt.pdf

[Editors' note: further revisions were suggested prior to acceptance, as described below.]

Thank you for resubmitting your work entitled "Limited column formation in the embryonic growth plate implies divergent growth mechanisms during pre- and postnatal bone development" for further consideration by *eLife*. Your revised article has been evaluated by Kathryn Cheah (Senior Editor), a Reviewing Editor, and two reviewers.

We would like to sincerely apologize for the substantial delay in getting back to you, which was due to the family emergency of both the reviewer and the reviewing editor. As you see below, we think that the additional concerns from the two reviewers can be addressed by modifying the main text/title and appending the discussion without any additional experimentation.

The manuscript has been improved but there are some remaining issues that need to be addressed, as outlined below:

*Reviewer #2 (Recommendations for the authors):*

First, I would like to apologize to the authors, the other reviewers, and the editor for this delayed response. I have been out of town dealing with medical issues pertaining to my parents. I should have been more responsive to the request ofr re-review, but each day's needs ate up all my time and energy.

I appreciate the effort by the authors to present new data and analyses to support the conclusions of the manuscript. I am especially pleased with the growth study as these data support a long-held conjecture that was never directly tested. I also appreciate the short-term labeling experiment, although these results raise more questions than answers as most small clones appear columnar and the unexpected large clones (not expected to exceed 4 cells in 48 hours) suggest either the presence of a subset of unusually proliferative cells or a potential caveat with the labeling method.

At this point, I support publication of these data with some small additions to the text. I am still not convinced of the conclusions, but I think the community should have access to the data to stimulate new experiments. The text additions are to put some of the findings into context and to highlight some remaining disconnects.

A. The claim that prenatal chondrocytes rarely form columns is at odds with two established observations in the field. (1) Histological analysis of tissue sections shows that most (80-90%) of groups of chondrocytes in the proliferative zone of embryonic cartilage exhibit what most would call columnar organization. I have revisited serial sections generated by my own lab to confirm this idea, which is also supported by the 2D images in this manuscript. (2) Live-imaging observations show that the majority (80-90%) of cell divisions result in rotation to a position consistent with column structure.

I think these differences between past literature and this manuscript should be explicitly stated and addressed in the discussion. What is the disconnect between these experimental observations?

B. I still don't understand why the cluster structure and orthogonal expansion is only observable by 3D clonal imaging and why cluster structures only observable from a "particular viewing angle". Regardless of how clones are oriented in the cartilage, sagittal and transverse histological or optical sections should intersect these cluster structures, but non-column structures are rarely observed in histological sections and many of your 2D images also predominantly show column structures. It is interesting that most of the non-columnar organizations and expansions appear in the z-axis (based on the representative images), for which resolution of optical microscopes is reduced by about 2.5 fold relative to the x and y axes. If clones are defined by cells labeled with the same color that are close together, close is different in the z-axis. What is unique about clusters such that only your method shows these cell groups?

C. In the discussion, you state "the very low rate of complete rotations we observed in the embryo suggests that the orientation of the division plane is tightly regulated to prevent such rotations." (1) You did not observe rotations. Rotations were inferred retrospectively from static data from fixed tissue. Only live imaging allows observation of rotations. This is a crucial distinction, and efforts should be made not to mislead the reader. (2) How is the orientation of the division plane the main factor in cluster formation? Live cell imaging established long ago that the division plane is tightly regulated in proliferative chondrocytes, most of which undergo normal rotations. It is rare to observe wild type chondrocytes that position the cell division plane (i.e., cleavage furrow) other than parallel (+/- 15 degrees) to the column axis to generate two cells that are initially oriented orthogonal to the column axis. How does regulating the orientation of the division plane factor into your model considering these established observations of cells that undergo normal rotation?

*Reviewer #3 (Recommendations for the authors):*

Many of my comments were consistent with those of Reviewers 1 and 2. I do appreciate that the authors changed the title of the manuscript, because I agree with Reviewer 2 that the first submission overstated the findings and their advance to the field. The new title is much more appropriate.

In Figure 5 S2, I appreciate the effort to correlate nuclear position with cell centroid, but I don't understand why the authors used previously published growth plate data for proximal tibia rather than segmenting the data in this manuscript to show this correlation holds at younger and older stages.

I do find Figure 1S2 to be helpful in thinking about the organization of these clones in three-dimensions. However, given these complex 3D morphologies, I think it's important to note in the second paragraph of the Results that these doublet pairs do not necessarily (and often don't) represent sister cells. I had previously thought these were sister pairs, so the complexity of cell orientations in a cluster was initially lost on me.

I appreciate the clone size subdivision to show that clone orientation isn't simply a function of clone size.

I appreciate that the authors recognize that there was previous awareness in the literature of the variance of columnar organization by species/growth plate/age, and I acknowledge that these past works were not able to take clonality into consideration. Yes, reviews and textbooks have oversimplified the illustration of 'growth plate organization', but this manuscript isn't the first data to show it's more nuanced, and although the prior literature wasn't so nicely quantitative, it was more than 'anecdotal'. Acknowledging these past works does not detract from a revised model with new data, and I think that Reviewer 2 and I both saw that this manuscript overstated its findings and impact on the field in this regard. I do appreciate that the authors have now added a sentence about the 'cell nests' observed by Dodd and that they discuss the fact that growth plate organization differs by species. The change in the title has helped, as did the removal of direct 'compare and contrast' with recent chick data.

I think that the quantification of elongation and lateral enlargement over time has added substantially to the manuscript and provides a plausible explanation for WHY there may be a difference in clonal organization in young versus older growth plates. This adds to the story in a meaningful way, and the revised manuscript is advances the field more than the initially submitted version.

Suggested title change: "Postnatal columnar organization of the mouse growth plate follows limited columnar organization in the embryo and shifts the axes of bone growth".

---

## [Author Response]

Essential revisions (for the authors):1) Prove the validity of the methodology that the authors used to measure nuclear elevation angles and define clonal orientations (morphometric analyses: are fetal clone sizes sufficiently large to reflect the orientation?) – related to comments by Reviewer 2 and 3.

The validation of our methodologies is described above in our responses to points 3 and 5 of Reviewer 2 and points 2-5 of Reviewer 3.

2) Establish the correlation between column/cluster organization and length-width ratios by calcein dye front labeling (or another method) to measure the change in the growth cartilage diameter over the same time – related to comments by Reviewer 1 and 2.

The correlation between column/cluster organization and length-width ratios is described in our responses to point 1 of Reviewer 1 and point 9 of Reviewer 3. These results were added to Figure 6 and supplementary figure 8 of the manuscript.

3) Stratify clone data according to the position in the resting and proliferative zones (top, middle, and bottom of each zone) to map clone development over time – related to comments by Reviewer 2.

As requested, we annotated the graphs and figures to mark the zones in which the clones belong.

4) Address inconsistencies with the literature and provide a more balanced consideration of the past findings to place the current findings better in context. Also, acknowledge the limitations of the prior and current analyses, and dampen/soften the conclusions/statements as appropriate – related to comments by all reviewers.

As requested, we have expanded the discussion on how our findings fit into the context of past discoveries. We also added a discussion on the limitations of past and current analyses, and softened our statements where relevant.

Reviewer #1 (Recommendations for the authors):With the shape data you already have in your group, you might be able to quantify the rates of longitudinal vs circumferential growth in the region studied in the present work. You would then be able to test the hypothesis that the change in orientation of the columns is correlated (or not) with the dominant direction(s) of growth over the time period of interest.

We thank Reviewers 1 and 3 for their suggestion to quantify the rates of longitudinal vs circumferential growth. Indeed, their suggestions were extremely helpful, leading to interesting discoveries about the relationship between elongation and expansion during bone growth. We quantified the rates of growth plate expansion and elongation in the distal femur, distal and proximal fibula, proximal humerus, distal radius, proximal and distal tibia, and distal ulna and found that the proportion of clusters and columns in these growth plates correlates with the ratio between expansion and elongation over time. These results were added to Figure 6 and supplementary figure 8.

It would be nice to make the code for the analysis pipeline available publicly in the interest of open science.

As requested, we have made the codes available publicly at the following link: https://github.com/ankitbioinfo/clonal_analysis_in_growth_plates_eLife

Reviewer #2 (Recommendations for the authors):Overall, this is an interesting study that introduces a more sophisticated clonal generation method and presents a new approach to data collection and analysis that together may also have high potential more generally for studies of tissue growth and morphology. However, there are several major issues that need addressing to increase confidence in these results and conclusions:1. The title "Bone elongation in the embryo occurs without column formation in the growth plate" is misleading for two reasons: (1) Figure 1 clearly shows columns of cells in the embryo – as acknowledged by the authors – and (2) the conclusion results from correlative studies and was not tested by experimental manipulation. One should not make such strong statements in the absence of mechanistic data or the presence of conflicting evidence. State what the data show, not what you want it to show.

As recommended by the reviewer, we changed the title to better reflect what our data show. The new title is “The ratio of clusters and columns in the growth plate is determined by modulation of division plane rotation in proliferating chondrocytes”.

2. The main argument in this manuscript appears to be more semantic than substantive. The authors report that the vast majority of clones in the embryo exist as clusters, yet 2D imaging of the embryo mostly reveals short columns (as stated above). The authors state that the cluster arrangement of clones can only be visualized in 3D. This suggests that "clusters" are possibly "clusters of columns" and not clusters of the structure diagrammed in Figure 6. As mentioned in the first paragraph, we already knew that not all cells rotate completely (some partial and some not) after division and therefore a single clone forming adjacent columns or "branching" is expected at some frequency based on the efficiency and geometry of rotation. It might be more accurate to state the current work shows that the efficiency of rotation is less or that under-rotating cells have a greater effect on clone morphology than previously thought. Either way, this is a refinement, not a violation, of the current mechanistic paradigm.

The definition of a column involves clonality, local cell stacking, and orientation of the clone in the growth plate. In the embryo, even if we ignore local cell stacking behaviors and focus only on global orientation of the clone with a permissive range of ±30 degrees, most clones do not orient in the direction of longitudinal bone growth. This violates the current mechanistic paradigm of a columnar structure and, thus, in our opinion the clones cannot be viewed as "clusters of columns". In the postnatal growth plate, where we did observe columns, we agree that branching does not change the paradigm, but rather is a refinement.

3. The emphasis on nuclear elevation measurements is not particularly convincing. Whether the alignment of nuclei reflects rotation or is simply the most efficient packing method for two cells in a constrained environment is not certain. Presenting the expectation of nuclear stacking is a "straw man" argument.

To determine whether nuclear centroids can be used as a proxy for cell centroids, we performed several analyses of the correlation between spatial relationships between nuclei and cells (Supplementary figure 6). For that, we used previously published growth plate data [1] of the proximal tibia, where cell and nuclear segmentations were available, to generate artificial clones. We then measured the elevation angles between cell doublets using either nuclear centroids or cell centroids as input. As seen in Supplementary figure 6A, the distributions of elevation angles were nearly identical. Additionally, Pearson correlation coefficient of 0.79 shows that the data based on nucleus and cell centroids are highly correlated (Supplementary figure 6B). Together, these results show that the alignment of nuclei reflects the amount of cell rotation.

4. Concerns about the measured data extend to the methodology. In general, the final "calculated" model should reflect what is observed in the raw data. The "clusters" described in the graphs or schematized in Figure 6 are not visible in the raw data. The authors mention the 2D vs. 3D problem, but this is not the issue. A single section (physical or optical) should cut through different clusters at different angles. The final result should be an image containing clone morphologies at the ratio in which they appear in the tissue. This is especially true for clones arranged medial to lateral – the predominant clone type reported – given the sagittal sections and radial symmetry of the growth plate. Unfortunately, the graphical data and the raw image data appear in conflict and therefore I do not have high confidence in this analysis and the resulting model.

To address the review’s concerns about the graphical model, we included in supplementary figure 2 several examples of embryonic clones from orthogonal views of the raw data. These images, as well as a collection of videos of clones, show that the graphical model reflects the raw image data. Additionally, all the raw data are available in a public repository at the following links:

https://doi.org/10.5281/zenodo.10440013, https://doi.org/10.5281/zenodo.10444731, https://doi.org/10.5281/zenodo.10446055, https://doi.org/10.5281/zenodo.10446092, https://doi.org/10.5281/zenodo.10446121, https://doi.org/10.5281/zenodo.10446131, https://doi.org/10.5281/zenodo.10446123, https://doi.org/10.5281/zenodo.10446145.

Finally, we added examples of clone morphologies to the model.

5. Although clonal analysis is a powerful technique, this specific study is predicated on the assumption that the clones observed in the proliferative zone originated from recombination events in single proliferative zone cells. However, a confounding factor is that endochondral ossification progressively removes proliferative chondrocytes through hypertrophy and remodeling of the calcified cartilage matrix. This means that the short (P-D axis) clones in the short proliferative zone of embryonic growth plates have limited resident lifetimes, which based on the growth rate is probably 24-48 hours. This would not affect the analysis of the longer clones of the postnatal growth plate that would require substantially longer periods to be completely removed. This is an important distinction because the authors induce recombination (cell labeling) in the embryo 96 hours before performing the analysis. Thus, most of the clones that were analyzed are likely to have derived from recombination events that occurred in the resting zone. Previous work demonstrated that resting zone clones expand in clusters and laterally rather than in columns as is common with proliferative chondrocytes. Therefore, a resting zone clone that enters the proliferative zone could appear as a single column (if labeled after the last resting cell division), or as a cluster of columns or a mixture of columns and other cell arrangements depending on how many cells in the resting clone are synchronously incorporated into the proliferative zone and how many cells in the new proliferative cluster divide before analysis. This is potentially a fatal flaw for this study.

We disagree with the reviewer’s assertion that allowing the cells to divide and rearrange for 4 days prior to analysis is a flaw. The growth plate is an engine that drives bone growth through the sequential morphogenesis and rearrangement of cells along its zones. The current dogma is that this happens through column formation and subsequent hypertrophy, which together directs growth along the longitudinal axis of the bone. Thus, the concept is that clone morphology supports growth in a preferred direction. If this is true, although we would expect most clones to originate in the proliferative zone, its origin does not matter. Additionally, the reviewer’s claim is based on the assumption that the proliferation rate and resident lifetime of chondrocytes are correlated. To our knowledge, no one has yet tested the behavior of proliferating chondrocytes experimentally and the proportion of chondrocytes that are stationary or move out of the zone is unknown.

To address the reviewer’s concerns about labeling time, we performed new pulsechase experiments of up to 48 hours (Author response image 1). The cell cycle in the proximal tibia growth plate is estimated at ~24 hours [2]. In agreement with this, we observed throughout the growth plate small clones composed primarily of two cells.

Interestingly, the few larger clones were elongated orthogonal clusters, similar to our previous results with the longer labeling time (Author response image 1 and G). Therefore, these results suggest that 96 hour labeling time is not the reason we rarely observed columns in the embryonic growth plate.

**Author response image 1. sa2fig1:** 48-hour labeling shows small clones that do not appear column-like. 3D morphology of chondrocyte clones was analyzed in the proximal tibia growth plate of *Col2a1CreER*^T2^:R26R-Confetti mice. (A) Cells were pulsed by tamoxifen administration at E15.5 and traced until E17.5. (B) Optical section through the growth plate shows sparse labeling throughout the PZ, PHZ, and HZ. (C) Zoomed-in optical PZ section shows small clones, some of which expand laterally. (D) 3D rendering of yfp, rfp, and cfp clones in the growth plate overlaid with nuclei in grey shows that most clones are single cells or doublets. (E) Highlight of two cfp clones from the PZ and PHZ. (F) Same region shown in D is rotated along the P-D axis. (G) Same region shown in in E shows that the clone long axis orients perpendicular to the P-D bone axis. Scale bars: B 100 µm, C 50 µm, D 70 µm, E 70 µm, F 70 µm, G 70 µm..

Reviewer #3 (Recommendations for the authors):

1. The authors compare their 'elevation angles' most directly to Li et al. *eLife* (2017). The two approaches, however, are entirely different. In Figure 1h of the 2017 paper, the authors describe that "For individual clones with more than two cells, the angle (<t) between the minor axis of each cell relative to the tissue proximodistal axis was measured (h1)." In this current manuscript under consideration, the authors describe the elevation angle as the angle between two nuclei in a spherical coordinate system (illustrated in Figure 1F). They are clear about describing these two approaches, but then they try to make a direct comparison by stating "Interestingly, perfect rotations characterized by elevation angles between 80-90°, which were previously predicted to be prevalent (19) were rare (. Figure 3G)." Reference 19 is Li et al. 2017. I don't think this is an equivalent comparison given the difference in measurement approach, cell axis versus nuclei position, further discussed below.

To address the reviewer’s concern, we removed the direct comparison of column orientations using the different methods. The revised sentence reads:

“Moreover, perfect rotations, characterized by elevation angles between 80°-90°, were rare (5.8% in the DF and 5.6% in the PT; Figure 3G).”

2. The authors also do not clearly describe how they measure the nuclear elevation angle. As described in the methods, they shifted the mean position of a doublet to the origin (0) in a spherical coordinate system, and then Figure 1F illustrates an elevation angle relative to an X-Z plane. It is unclear from this illustration and from the description in the methods exactly how the XZ plane is established. It is also unclear from this illustration exactly how the elevation angle is calculated in three dimensions given that the three schematics (0, 45, 90 degrees) differ only in a two dimensional plane. There are examples for real cells in Figure 1G, but the figure resolution makes it impossible to tell which are solid and which are dashed black lines and what is the relevant angle.

We clarified in the Methods section and in Figure 1F and its legend how the elevation angle was calculated. Additionally, we added to the Results section a citation of a recent preprint that uses the same method to calculate elevation angles between cilia and nuclei [3]. The revised Methods subsection is listed below:

To determine the elevation angle between nuclei in a doublet, we first transformed the Cartesian coordinates of doublet to a spherical coordinate system, in which the Z-axis represents the P-D bone axis. Then, we shifted the mean position of a doublet to the origin zero and used the MATLAB function cart2sph to obtain the elevation angle (phi) and the radius (r).

The elevation angle was measured as the angle between the projected line of a nucleus on the XY plane to the line connecting two nuclear centroids (Figure 1F). The XY plane is perpendicular to the Z-axis. If one nucleus in the doublet has an elevation angle of phi, the angle of the other nucleus is -phi. The elevation angle values are in the range of [-pi/2, pi/2]. The radius in the spherical coordinate system is equivalent to half of the distance between two nuclei in a Cartesian coordinate system. The elevation angle between two nuclei lying on top of one another is 90°, whereas the angle between two nuclei that are positioned next to each other in the XY plane is 0. Because the elevation angles of nuclei in a doublet mirror each other, we only used the absolute value.

3. Perhaps using the geometric center of each cell rather than the nuclei positions would be a more accurate reflection of cell position.

Similar to our response to point 3 of Reviewer 2, we addressed this request experimentally by performing correlation analysis between elevation angles calculated from cell or nucleus centroids, as shown in Supplementary figure 6. Our results indicate that nuclear centers are a good proxy for cell centers.

4. The methods state "Clone morphometrics, such as PC coefficients and PC orientations, were calculated as described previously (7)." The referenced manuscript describes an approach to extract axes for individual cells but not for more complex clonal morphologies. Does PC1 map through the furthest two points on the perimeter of a clone, which is what I could consider the 'long axis' of a clone? If not, why not and what is the justification for using the PC1 axis for clone orientation versus the furthest two points on a perimeter?

PC1 maps the first principal component of the clone mask, corresponding to the major axis of the clone, along which the point clouds have the highest variance. The clone mask represents the combined cell cytoplasm and, thus, PC1 represents the long axis of the clone. We overlaid the PC1 vector with the clone mask and observed that it faithfully represents the long axis of the clone in both clusters and columns. Author response image 2 shows several examples of embryonic and postnatal clones with their PC1 vectors indicated by black arrows.

**Author response image 2. sa2fig2:** PC1 vectors overlaid on clones. Three orthogonal viewing angles (XZ, YZ, XY) of PC1 vectors marked by black arrow overlaid on four clones from embryonic (a-d) and postnatal (e-h) growth plates. a,b,e,f are clusters, whereas c,d,g,h are columns. Z-axis represents the P-D bone axis. Clone masks representing cell cytoplasm are in magenta, nuclei are in cyan.

5. The postnatal labeling time was 10 days, and clone sizes were **much** larger postnatally than at the fetal time point. This gives opportunity for larger clones to develop a longer long axis, which may align better with the true axis of growth. By contrast, the labeling duration of 4 days from E14.5 may not achieve sufficient clone sizes to reflect the overall proliferative zone growth structure. For example, if two flat cells with a long axis diameter of 10 μm each sit side by side in a small clone within a column that is aligned with the growth axis, there would have to be more than four well stacked cells with a short axis diameter of 5 μm each for the clone long axis to reflect its growth axis. Therefore, smaller clones could have axes that aren't reflective of the larger order they might be contained within. Either the labeling time should be extended, or the authors should show that fetal clone sizes are sufficiently large to reflect the orientation of their expansion.

To address the reviewer’s concerns about the labeling time, we compared the ratio of columns per clone size in pre- and postnatal growth plates (Author response image 3). When we compared clones of the same size, i.e. 3-5 cells, 6-10 cells, and 11-20 cells, columns were consistently more frequent in postnatal growth plates. This suggests that the ability of cells to form columns is innately different between these two developmental stages. Additionally, we were fortunate to receive raw imaging data from Newton *et al.* (2019), who performed a 10-day pulse and chase experiment (E14.5 – P5) using the same mouse strains. Similar to our results, the clones in these growth plates also failed to form columns.

**Author response image 3. sa2fig3:** Column frequency per clone size. Frequency of columns and clusters as a function of clone size in embryonic (A,B) and postnatal growth plates (C,D). When comparing clones of the same size, columns are more frequent postnatally.

6. Interpretations – that this is a 'discovery' that some growth plates have no columns or that columns aren't perfectly organized in stacks and that 'longitudinal bone growth is governed by different cellular mechanisms during embryonic and postnatal development'. It has been documented that there is great variance in growth plate morphology both between species, within species, and by age. This includes variance in proliferative zone 'columnar organization'. The differences have been associated with relative growth rate with defined columns and larger hypertrophic chondrocytes observed in faster growing bones [see for example Dodds *The Anatomical Record* (1930), Wilsman et al. *Journal of Orthopaedic Research* (2008), and Hunziker *Microscopy Research and Technique* (1994)]. Dodds notes “'cell nests' in the distal end of a human infant phalanx (Figure 13 below) that 'do not pass through the flattened stage and no rows are formed' prior to enlarging near the marrow front and states that 'in the proximal end of the same bone the cells are in typical rows' (Figure 12 below). Therefore, a lack of strict columnar organization in growth plate cartilage is not a new observation.”

We are aware of the differential growth of growth plates, which is why we studied columns in the fast-growing growth plates of the proximal tibia and distal femur [2, 4-7]. The papers referenced by the reviewer do not report results of lineage tracing analyses, without which it is impossible to determine whether or not cell clones form columns. Clonality is a key part of the column definition. Additionally, while possible variability in column formation may have been discussed anecdotally, this is not reflected in the textbooks. As this is the first rigorous lineage tracing-based 3D study of column formation, we stand by our statement that the scarcity of clonal columns in the embryonic growth plate (less than 5%) is a discovery. To put our discovery in context, we added to the Discussion the limitations that we cannot rule out the possibility that clonal columns exist in other species.

7. Ernst Hunziker also notes the striking differences between species in his 1994 review. He remarks that "The avian growth plate (proximal tibia; 4-week-old chick) is characterized by a very high cellularity, which somewhat masks a fairly high degree of structural anisotropy." It is therefore not surprising that the observations here in mice might differ from observations in chicken embryos (Li et al. 2017) even if the quantifications of cell orientations were done the same.

We agree with the reviewer that we cannot rule out the possibility that our observations are unique to mice. To prevent confusion regarding the reach of our claims, we added to the Discussion a paragraph about possible diversity in growth plates of different organisms.

8. Most of the small orthogonally oriented clusters are clearly within the resting zone or very upper proliferative zone whereas the elongate clones are in the proliferative zone, and the authors state this. Dodds also described these 'cell nests' within the resting zone (epiphyseal) as small clusters separated by matrix space. Their discovery in this analysis is not new or surprising.

To clarify that “cell nests” were previously described as small clusters, we added a reference to Dodds’ paper to the Discussion.

9. Specifically, I would like to see the authors measure the rate of linear growth in each stage [P14.5-18.5 (or longer) and P30-P40], by calcein dye front labeling or another method, and to measure the change in diameter of the growth cartilage over the same time. I suspect that the ratio of longitudinal/axial growth at the fetal stage is lower than at the late juvenile stage, consistent with the difference in clone orientation orthogonal to the elongation axis. This would support the interpretation proposed in the discussion that clones oriented more orthogonal to the long axis might support greater axial versus longitudinal growth.

This very helpful suggestion was also made by Reviewer 1 (point 1). As detailed in our response to that comment, we performed this analysis, which indeed produced interesting findings on the relationship between the proportion of clusters and columns and elongation and expansion during bone growth. These results were added to Figure 6 and supplementary figure 8 (Figures L1 and L2 above).

10. In conclusion, I think the methods of measuring cell stacking and clone orientation need to be more clearly described and justified, especially to account for differences in nuclear versus cell stacking.

As mentioned, we clarified in the Methods section and in Figure 1F how the elevation angle was calculated and added a citation of a recent preprint reporting a similar analysis of elevation angles between cilia and nuclei. We performed correlation analysis to show that nuclear centers are a good proxy for cell centers for computation of elevation angles (Supplementary figure 6). Lastly, in the reply to point 4 above, we provide justification for clone orientation measurements.

11. I also think the findings need to be placed more accurately in the context of existing literature, because they are not as novel and surprising as they are currently presented.

As requested by the reviewer, we have expanded the discussion on how our findings fit into the context of existing literature.

12. Instances of 'embryonic' should be replaced with 'fetal', and these clones were labeled from E14.5 to E18.5. These are fetal stages.

The distinction between embryonic and fetal stages is important in the context of human in utero development. However, in studies of mouse development, this distinction is rarely made and the terms “embryo” and “embryonic” are used throughout gestation. For example, this citation is from a review by Crawford et al. (2010), which also references the seminal “The Atlas of Mouse Development” by Matthew H. Kaufman:

“The term “embryo” is used for the developing human individual from the time of implantation until the time of onset of bone marrow formation in the humerus, which is about the end of the eighth week postconception. After this stage and until birth, the term “fetus” is used… Since the mouse has a much shorter gestation period, the designation of “embryo” versus “fetus” is less important, whereas the developmental age post-conception is critically important. For this reason, the term “embryo” is used to define all stages of murine development between fertilization and birth with the stage of development indicated by the gestational age (E0.5 + days postconception) (Kaufman 1999).”

Crawford LW, Foley JF, Elmore SA. Histology atlas of the developing mouse hepatobiliary system with emphasis on embryonic days 9.5-18.5. Toxicol Pathol. 2010 Oct;38(6):872-906.

13. In the introduction "Most extreme is the resting zone." The use of 'extreme' here is unclear.

Perhaps instead 'most distal with respect to the diaphyseal bone'

As requested, we replaced the phrase “Most extreme” with “At the most distal epiphyseal end”.

14. "Columnar arrangement facilitates bone elongation by maximizing *cell density* in the longitudinal axis while limiting it laterally." What is meant here by 'cell density'? Cells per area of tissue? I don't think this has been quantified, and it would vary by species/stage and matrix thickness. I think the authors mean that more cells are passing through the longitudinal axis, so perhaps instead use 'cell turnover'?

What we meant by this quote is that cells in the growth plate are concentrated along the proximal-distal axis as opposed to the medial-lateral or dorsal-ventral axis. We prefer to keep this phrasing as was proposed in Romereim S.M. *et al.*, (2011)*.*

15 Note in the Figure legends that these are 200 μm thick sections, and also note the average number of cells that fit within that Z thickness

As requested, we added to the Methods section the mean number of cells that fits within a 200-micrometer Z thickness.

References

Rubin, S., et al., Application of 3D MAPs pipeline identifies the morphological sequence chondrocytes undergo and the regulatory role of GDF5 in this process. Nature Communications, 2021. 12(1): p. 5363.

Wilsman, N.J., et al., Cell cycle analysis of proliferative zone chondrocytes in growth plates elongating at different rates. Journal of Orthopaedic Research, 1996. 14(4): p. 562572.

Carolyn, M.O., et al., Nanometer-scale views of visual cortex reveal anatomical features of primary cilia poised to detect synaptic spillover. bioRxiv, 2023: p. 2023.10.31.564838.

Stern, T., et al., Isometric Scaling in Developing Long Bones Is Achieved by an Optimal Epiphyseal Growth Balance. PLoS biology, 2015. 13(8): p. e1002212-e1002212.

Wilsman, N.J., et al., Differential growth by growth plates as a function of multiple parameters of chondrocytic kinetics. Journal of Orthopaedic Research, 1996. 14(6): p. 927-936.

Wilsman, N.J., et al., Age and pattern of the onset of differential growth among growth plates in rats. Journal of orthopaedic research : official publication of the Orthopaedic Research Society, 2008. 26(11): p. 1457-1465.

Lui, J.C., et al., Differential aging of growth plate cartilage underlies differences in bone length and thus helps determine skeletal proportions. PLoS biology, 2018. 16(7): p. e2005263-e2005263.

Crawford LW, Foley JF, Elmore SA. Histology atlas of the developing mouse hepatobiliary system with emphasis on embryonic days 9.5-18.5. Toxicol Pathol. 2010 Oct;38(6):872-906.

Romereim SM, Conoan NH, Chen B, Dudley AT. A dynamic cell adhesion surface regulates tissue architecture in growth plate cartilage. Development. 2014 May;141(10):2085-95. doi: 10.1242/dev.105452. Epub 2014 Apr 24. PMID: 24764078; PMCID: PMC4011088.

Rubin, S., et al., Application of 3D MAPs pipeline identifies the morphological sequence chondrocytes undergo and the regulatory role of GDF5 in this process. Nature Communications, 2021. 12(1): p. 5363.

Wilsman, N.J., et al., Cell cycle analysis of proliferative zone chondrocytes in growth plates elongating at different rates. Journal of Orthopaedic Research, 1996. 14(4): p. 562572.

Carolyn, M.O., et al., Nanometer-scale views of visual cortex reveal anatomical features of primary cilia poised to detect synaptic spillover. bioRxiv, 2023: p. 2023.10.31.564838.

Stern, T., et al., Isometric Scaling in Developing Long Bones Is Achieved by an Optimal Epiphyseal Growth Balance. PLoS biology, 2015. 13(8): p. e1002212-e1002212.

Wilsman, N.J., et al., Differential growth by growth plates as a function of multiple parameters of chondrocytic kinetics. Journal of Orthopaedic Research, 1996. 14(6): p. 927-936.

Wilsman, N.J., et al., Age and pattern of the onset of differential growth among growth plates in rats. Journal of orthopaedic research : official publication of the Orthopaedic Research Society, 2008. 26(11): p. 1457-1465.

Lui, J.C., et al., Differential aging of growth plate cartilage underlies differences in bone length and thus helps determine skeletal proportions. PLoS biology, 2018. 16(7): p. e2005263-e2005263.

Crawford LW, Foley JF, Elmore SA. Histology atlas of the developing mouse hepatobiliary system with emphasis on embryonic days 9.5-18.5. Toxicol Pathol. 2010 Oct;38(6):872-906.

Romereim SM, Conoan NH, Chen B, Dudley AT. A dynamic cell adhesion surface regulates tissue architecture in growth plate cartilage. Development. 2014 May;141(10):2085-95. doi: 10.1242/dev.105452. Epub 2014 Apr 24. PMID: 24764078; PMCID: PMC4011088.

[Editors’ note: what follows is the authors’ response to the second round of review.]

The manuscript has been improved but there are some remaining issues that need to be addressed, as outlined below:Reviewer #2 (Recommendations for the authors):First, I would like to apologize to the authors, the other reviewers, and the editor for this delayed response. I have been out of town dealing with medical issues pertaining to my parents. I should have been more responsive to the request ofr re-review, but each day's needs ate up all my time and energy.I appreciate the effort by the authors to present new data and analyses to support the conclusions of the manuscript. I am especially pleased with the growth study as these data support a long-held conjecture that was never directly tested. I also appreciate the short-term labeling experiment, although these results raise more questions than answers as most small clones appear columnar and the unexpected large clones (not expected to exceed 4 cells in 48 hours) suggest either the presence of a subset of unusually proliferative cells or a potential caveat with the labeling method.At this point, I support publication of these data with some small additions to the text. I am still not convinced of the conclusions, but I think the community should have access to the data to stimulate new experiments. The text additions are to put some of the findings into context and to highlight some remaining disconnects.A. The claim that prenatal chondrocytes rarely form columns is at odds with two established observations in the field. (1) Histological analysis of tissue sections shows that most (80-90%) of groups of chondrocytes in the proliferative zone of embryonic cartilage exhibit what most would call columnar organization. I have revisited serial sections generated by my own lab to confirm this idea, which is also supported by the 2D images in this manuscript. (2) Live-imaging observations show that the majority (80-90%) of cell divisions result in rotation to a position consistent with column structure.I think these differences between past literature and this manuscript should be explicitly stated and addressed in the discussion. What is the disconnect between these experimental observations?

To address the differences highlighted by the reviewer between past literature and our manuscript we have added a paragraph to the discussion that explains these apparent differences.

“Our discovery that during embryogenesis, when bone elongation is at its highest, the growth plates contain only few columns, contradicts previous studies and reveals the need to reconsider the underlying mechanisms. Several possibilities could explain these differences. One plausible explanation is that earlier studies focused on the mechanism of division plane rotation in column formation, but they either analyzed small subsets of clonal doublets without tracking their contributions to columns (*19, 20*) or examined non-clonal doublets (*30*). Without integrating these aspects into a comprehensive study, it is difficult to confirm the existence of clonal columns and determine whether they are formed by cells that have undergone complete division plane rotation. Additionally, it is plausible that the clusters we identified intercalate in a manner that forms geometric non-clonal columns. This could explain the columnar arrangement of chondrocytes observed in histological sections of the growth plate.”

B. I still don't understand why the cluster structure and orthogonal expansion is only observable by 3D clonal imaging and why cluster structures only observable from a "particular viewing angle". Regardless of how clones are oriented in the cartilage, sagittal and transverse histological or optical sections should intersect these cluster structures, but non-column structures are rarely observed in histological sections and many of your 2D images also predominantly show column structures. It is interesting that most of the non-columnar organizations and expansions appear in the z-axis (based on the representative images), for which resolution of optical microscopes is reduced by about 2.5 fold relative to the x and y axes. If clones are defined by cells labeled with the same color that are close together, close is different in the z-axis. What is unique about clusters such that only your method shows these cell groups?

The reviewer raised the possibility that our results might be due to imaging artifacts. To address this concern, we would like to draw the reviewer's attention to Figure 1 —figure supplement 2 and Videos 1-11, where it is clearly apparent that the clusters are visible in all axes, not just along the Z axis. Furthermore, as cited in our paper, the study by Decker et al. (2017) on articular cartilage found that upon chondrocyte proliferation, clusters and stacks of non-daughter cells were formed. This provides additional support that our observations are not artifacts. Finally, to rule out any possibility that our data is flawed in a way that could lead to incorrect conclusions, we reached out to Prof. Andrei S. Chagin, who generously shared with us his imaging data from Newton et al. (2019). In this study, they initiated labeling of chondrocytes in the growth plate at E14.5 and analyzed clones at P5. As shown in Author response image 3 of our last response letter, we found that, similar to our data, many clones form clusters rather than columns.

C. In the discussion, you state "the very low rate of complete rotations we observed in the embryo suggests that the orientation of the division plane is tightly regulated to prevent such rotations." (1) You did not observe rotations. Rotations were inferred retrospectively from static data from fixed tissue. Only live imaging allows observation of rotations. This is a crucial distinction, and efforts should be made not to mislead the reader. (2) How is the orientation of the division plane the main factor in cluster formation? Live cell imaging established long ago that the division plane is tightly regulated in proliferative chondrocytes, most of which undergo normal rotations. It is rare to observe wild type chondrocytes that position the cell division plane (i.e., cleavage furrow) other than parallel (+/- 15 degrees) to the column axis to generate two cells that are initially oriented orthogonal to the column axis. How does regulating the orientation of the division plane factor into your model considering these established observations of cells that undergo normal rotation?

We agree with the reviewer's suggestion and have changed the wording of this paragraph.

“The rotation of the division plane is the cellular mechanism that underlies the formation of columns versus clusters. If indeed these cellular modalities are utilized to promote different growth strategies, then the mechanism that controls the division plane can be central in the regulation of bone morphology. Moreover, it can provide an adaptable means to shift between different growth strategies such as elongation, expansion, and curvature formation. Interestingly, a simulation of cell adhesion in the growth plate showed that considering the mechanical confinement of the chondrocyte matrix, a complete rotation is energetically favorable (*44*). Thus, the low rates of complete division plane rotations, deduced from the doublet elevation angle analysis, suggests that the orientation of the division plane is tightly regulated.”

Regarding the further questions raised by the reviewer, we have provided a comprehensive answer, which we have now included in the Discussion section in response to the first remark. We have also included in the discussion a paragraph stating the limitations of our study.

“One limitation of our rotation analysis is that in order to calculate the relative position between neighboring cells, we assumed that chondrocyte rotation and orientation are correlated and that chondrocytes do not move after rotating, as was previously suggested (20, 30)”

Finally, assuming that after division the chondrocytes do not change their positions, then in cases where the rotation of the division plane is partial, instead of stacking to form a column, the cells would spread horizontally, forming a cluster.

Reviewer #3 (Recommendations for the authors):Many of my comments were consistent with those of Reviewers 1 and 2. I do appreciate that the authors changed the title of the manuscript, because I agree with Reviewer 2 that the first submission overstated the findings and their advance to the field. The new title is much more appropriate.In Figure 5 S2, I appreciate the effort to correlate nuclear position with cell centroid, but I don't understand why the authors used previously published growth plate data for proximal tibia rather than segmenting the data in this manuscript to show this correlation holds at younger and older stages.

Accurate 3D cell segmentation is a time-consuming process. Since we already had a large database of cell and nucleus pairs segmented from E16.5 growth plates, we utilized this data for the cell-nucleus correlation study. We have no reason to suspect any variation in the behavior of the cells and their nuclei between E16.5 and E18.5.

I do find Figure 1S2 to be helpful in thinking about the organization of these clones in three-dimensions. However, given these complex 3D morphologies, I think it's important to note in the second paragraph of the Results that these doublet pairs do not necessarily (and often don't) represent sister cells. I had previously thought these were sister pairs, so the complexity of cell orientations in a cluster was initially lost on me.

We agree with the reviewer on the need to clarify that we measure all doublet pairs and not only those which represent sister cells. We changed the sentence to the following:

“For that, we measured the elevation angle between all neighboring pairs of cells (doublets) within a given clone, similar to what was done previously (*40*) (see details in Methods and Figure 1F).”

[…]Suggested title change: "Postnatal columnar organization of the mouse growth plate follows limited columnar organization in the embryo and shifts the axes of bone growth".

We believe that our current title “Limited column formation in the embryonic growth plate implies divergent growth mechanisms during pre- and postnatal bone development” better describes the importance of our findings.